# Robustness of Deep Learning for Accelerated MRI: Benefits of Diverse Training Data

## Abstract

Deep learning based methods for image reconstruction are state-of-the-art for a variety of imaging tasks. However, neural networks often perform worse if the training data differs significantly from the data they are applied to. For example, a network trained for accelerated magnetic resonance imaging (MRI) on one scanner performs worse on another scanner. In this work, we investigate the impact of the training data on the model's performance and robustness for accelerated MRI. We find that models trained on the combination of various data distributions, such as those obtained from different MRI scanners and anatomies, exhibit robustness equal or superior to models trained on the best single distribution for a specific distributions shift. Thus training on diverse data tends to improve robustness. Furthermore, training on diverse data does not compromise in-distribution performance, i.e., a model trained on diverse data yields in-distribution performance at least as good as models trained on the more narrow individual distributions. Our results suggest that training a model for imaging on a variety of distributions tends to yield a more effective and robust model than maintaining separate models for individual distributions.

## 1 Introduction

Deep learning models trained end-to-end for image reconstruction are fast and accurate and outperform traditional image reconstruction methods for a variety of imaging tasks ranging from denoising over super-resolution to accelerated MRI (Jin et al., 2017; Liang et al., 2021; Dong et al., 2014; Muckley et al., 2021). Imaging accuracy is typically measured as in-distribution performance: A model trained on data from one source is applied to data from the same source.

However, in practice a neural network for imaging is typically applied to slightly different data than it is trained on. For example, a neural network for accelerated magnetic resonance imaging trained on data from one hospital is applied in a different hospital.

Neural networks for imaging often perform significantly worse under such distribution shifts. For accelerated MRI, a network trained on knee images performs worse on brain images when compared to the same network trained on brain images, and similar performance loss occurs for other natural distribution shifts (Knoll et al., 2019; Johnson et al., 2021; Darestani et al., 2021).

To date, much of research in deep learning for imaging has focused on developing better models and algorithms to improve *in-distribution* performance. Nevertheless, recent literature on *computer vision* models, in particular multi-modal models, suggest that a model's robustness is largely impacted by the training data, and a key ingredient for robust models are large and diverse training sets (Fang et al., 2022; Nguyen et al., 2022; Gadre et al., 2023).

In this paper, we take a step towards a better understanding of the training data for learning robust deep networks for accelerated magnetic resonance imaging (MRI).

First, we investigate whether deep networks for accelerated MRI compromise performance on individual distributions when trained on more than one distribution. We find for various pairs of distributions (different anatomies, image contrasts, and magnetic fields), training a single model on two distributions yields the same performance as training two individual models.

Second, we demonstrate for a variety of distribution shifts (anatomy shift, image contrast shift, and magnetic field shift) that the robustness of models, regardless of its architecture, is largely determined by the training set and a diverse set enhances robustness towards distribution shifts.

Third, we consider a distribution shift from healthy to non-healthy subjects and find that models trained on a diverse set of healthy subjects can accurately reconstruct images with pathologies even if the model has never seen pathologies during training.

Fourth, we empirically find for a variety of distribution shifts that *distributional* overfitting occurs: When training for long, in-distribution performance continues to improve slightly while out-of-distribution performance sharply drops. A related observation was made by Wortsman et al. (2022) for fine-tuning of CLIP models. Therefore, early stopping can be helpful for training a robust model as it can yield a model with almost optimal in-distribution performance without losing robustness.

Taken together, those four findings suggest that training a single model on a diverse set of data distributions and incorporating early stopping yields a robust model. We test this hypothesis by training a model on a large and diverse pool of data significantly larger than the fastMRI dataset, the single largest dataset for accelerated MRI, and find that the network is significantly more robust than a model trained on the fastMRI dataset, without compromising in-distribution performance.

**Related Work.** A variety of influential papers have shown that machine learning methods for problems ranging from image classification to natural language processing perform worse under distribution shifts (Recht et al., 2019; Miller et al., 2020; Taori et al., 2020; Hendrycks et al., 2021).

With regards to accelerated MRI Johnson et al. (2021) evaluate the out-of-distribution robustness of the models submitted to the 2019 fastMRI challenge (Knoll et al., 2020), and find that they are sensitive to distribution shifts. Furthermore, Darestani et al. (2021) demonstrate that reconstruction methods for MRI, regardless of whether they are trained or only tuned on data, all exhibit similar performance loss under distribution shifts. Both work do not propose robustness enhancing strategies, such as training on a diverse dataset. Moreover, there are several works that characterise the severity of specific distribution-shifts and propose transfer learning as a mitigation strategy (Knoll et al., 2019; Huang et al., 2022; Dar et al., 2020). Those works fine-tune on data from the test distribution, whereas we study-out-of-distribution setup without access to the test distribution.

A potential solution to enhance robustness in accelerated MRI is offered by Darestani et al. (2022), who introduce test-time training to narrow the performance gap on out-of-distribution data, albeit with high computational costs. In the context of ultrasound imaging, Khun Jush et al. (2023) demonstrate that diversifying simulated training data can improve robustness on real-world data. Liu et al. (2021) propose a special network architecture to improve the performance of training on multiple anatomies simultaneously. Ouyang et al. (2023) proposes an approach that modifies natural images for training MRI reconstruction models.

Shifting to computer vision, OpenAI's CLIP model (Radford et al., 2021) is remarkably robust under distribution shifts. Fang et al. (2022) explore this finding and show that the key contributor to CLIP's robustness is the diversity training. However, Nguyen et al. (2022) show that blindly combining data can weaken robustness compared to training on the best individual data source.

These studies underscore the pivotal role of dataset design, particularly data diversity, for a model's performance and robustness. In light of concerns regarding the robustness of deep learning in medical imaging, we explore the impact of data diversity on models trained for accelerated MRI.

## 2   SETUP AND BACKGROUND

**Reconstruction task.** We consider multi-coil accelerated MRI, where the goal is to reconstruct a complex-valued, two-dimensional image $\mathbf{x} \in \mathbb{C}^N$ from the measurements of electromagnetic signals obtained through $C$ receiver coils according to

$$\mathbf{y}_i = \mathbf{MFS}_i\mathbf{x} + \mathbf{z}_i \in \mathbb{C}^m, \quad i = 1, \ldots, C. \tag{1}$$

Here, $\mathbf{x} \in \mathbb{C}^N$ is the image to be reconstructed, $\mathbf{S}_i$ is the sensitivity map of the $i$-th coil, $\mathbf{F}$ is the 2D discrete Fourier transform, $\mathbf{M}$ is an undersampling mask, and $\mathbf{z}_i$ models additive white Gaussian noise. The measurements $\mathbf{y}_i$ are often called k-space measurements (see illustration in Appendix A).

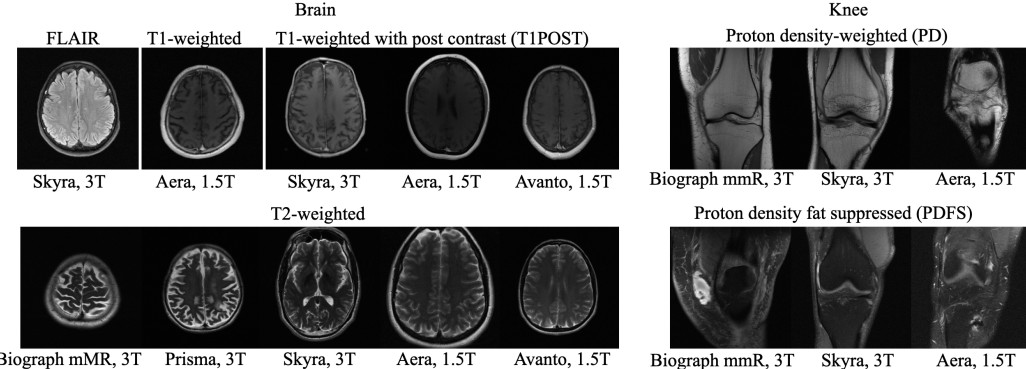

Figure 1: Randomly selected example images for a selection of distributions from the fastMRI dataset (Zbontar et al., 2019) we consider here. Axial view brain images are on the left, coronal view knee images are on the right. The caption above an image describes the image contrast, and the caption below is the name of the MRI scanner used.

Table 1: Fully-sampled k-space datasets used here. The percentages are the proportions of the data within a dataset. Scans containing multiple echoes or averages are separated as such and counted as separate volumes.

| Dataset | Anatomy | View | Image contrast | Vendor | Magnet | Coils | Vol./Subj. | Slices |
|---|---|---|---|---|---|---|---|---|
| fastMRI knee (Zbontar et al., 2019) | knee | coronal | PD (50%), PDFS (50%) | Siemens | 1.5T (45%), 3T (55%) | 15 | 1.2k/1.2k | 42k |
| fastMRI brain (Zbontar et al., 2019) | brain | axial | T1 (11%), T1POST (21%), T2 (60%), FLAIR (8%) | Siemens | 1.5T (43%), 3T (67%) | 4-20 | 6.4k/6.4k | 100k |
| fastMRI prostate (Tibrewala et al., 2023) | prostate | axial | T2 | Siemens | 3T | 10-30 | 312/312 | 9.5k |
| M4Raw (Lyu et al., 2023) | brain | axial | T1 (37%), T2 (37%), FLAIR (26%) | XGY | 0.3T | 4 | 1.4k/183 | 25k |
| SKM-TEA, 3D (Desai et al., 2021) | knee | sagittal | qDESS | GE | 3T | 8, 16 | 310/155 | 50k |
| Stanford 3D (Epperson, 2013) | knee | axial | PDFS | GE | 3T | 8 | 19/19 | 6k |
| Stanford 3D (Epperson, 2013) | knee | coronal | PDFS | GE | 3T | 8 | 19/19 | 6k |
| Stanford 3D (Epperson, 2013) | knee | sagittal | PDFS | GE | 3T | 8 | 19/19 | 4.8k |
| 7T database, 3D (Caan, 2022) | brain | axial | MP2RAGE-ME | Philips | 7T | 32 | 385/77 | 112k |
| 7T database, 3D (Caan, 2022) | brain | coronal | MP2RAGE-ME | Philips | 7T | 32 | 385/77 | 112k |
| 7T database, 3D (Caan, 2022) | brain | sagittal | MP2RAGE-ME | Philips | 7T | 32 | 385/77 | 91k |
| CC-359, 3D (Souza et al., 2018) | brain | axial | GRE | GE | 3T | 12 | 67/67 | 17k |
| CC-359, 3D (Souza et al., 2018) | brain | coronal | GRE | GE | 3T | 12 | 67/67 | 14k |
| CC-359, 3D (Souza et al., 2018) | brain | sagittal | GRE | GE | 3T | 12 | 67/67 | 11k |
| Stanford 2D (Cheng, 2018) | various | various | various | GE | 3T | 3-32 | 89/89 | 2k |
| NYU data (Hammernik et al., 2018) | knee | various | PD (40%), PDFS (20%), T2FS(40%) | Siemens | 3T | 15 | 100/20 | 3.5k |
| M4Raw GRE (Lyu et al., 2023) | brain | axial | GRE | XGY | 0.3T | 4 | 366/183 | 6.6k |

In this work, we consider 4-fold accelerated (i.e., $m = N/4$) multi-coil 2D MRI reconstruction with 1D Cartesian undersampling. The central k-space region is fully sampled including 8% of all k-space lines, and the remaining lines are sampled equidistantly with a random offset from the start. We choose 4-fold acceleration as going beyond 4-fold acceleration, radiologists tend to reject the reconstructions by neural networks (Muckley et al., 2021; Radmanesh et al., 2022). Equidistant sampling is chosen due to the ease of implementation on existing machines (Zbontar et al., 2019).

**Class of reconstruction methods.** We focus on deep learning models trained *end-to-end* for accelerated MRI, as this class of methods consistently deliver state-of-the-art performance in accuracy and speed (Hammernik et al., 2018; Aggarwal et al., 2019; Sriram et al., 2020; Fabian et al., 2022). A neural network $f_{\boldsymbol{\theta}}$ with parameters $\boldsymbol{\theta}$ mapping a measurement $\mathbf{y} = \{\mathbf{y}_1, \ldots, \mathbf{y}_C\}$ to an image is most commonly trained to reconstruct an image from the measurements $\mathbf{y}$ by minimizing the supervised loss $\mathcal{L}(\boldsymbol{\theta}) = \sum_{i=1}^{n} \text{loss}(f_{\boldsymbol{\theta}}(\mathbf{y}_i), \mathbf{x}_i)$ over a training set consisting of target image and corresponding measurements $\{(\mathbf{x}_1, \mathbf{y}_1), \ldots, (\mathbf{x}_n, \mathbf{y}_n)\}$. This dataset is typically derived from fully-sampled k-space data (i.e., data where the undersampling mask $M$ is identity). From the fully-sampled k-space data, a target image $\mathbf{x}$ is estimated, and retrospectively undersampled measurements $\mathbf{y}$ are generated by applying the undersampling mask to the fully-sampled data.

Several choices of network architectures work well. A standard baseline is a U-net (Ronneberger et al., 2015) trained to reconstruct the image from a coarse least-squares reconstruction of the measurements (Zbontar et al., 2019). A vision transformer for image reconstruction applied in the same fashion as the U-net also works well (Lin & Heckel, 2022). The best-performing models are un-rolled neural networks such as the variational network (Hammernik et al., 2018) and a deep cascade

of convolutional neural networks (Schlemper et al., 2018). The unrolled networks often use either the U-net as backbone, like the end-to-end VarNet (Sriram et al., 2020), or a transformer based architecture (Fabian et al., 2022).

We expect our results in this paper to be model agnostic, as related works have demonstrated that factors like robustness and changes in the training set affect a large variety of different model architectures equally (Darestani & Heckel, 2021; Miller et al., 2021). We show that this is indeed the case for convolutional, transformer, and unrolled networks.

**Datasets.** We consider the datasets listed in Table 1, which are all fully-sampled MRI dataset with varying attributes. The datasets include the largest publicly available fully-sampled MRI datasets.

Most of our experiments are based on splits of the fastMRI dataset (Zbontar et al., 2019), the most commonly used dataset for MRI reconstruction research. Figure 1 depicts samples from the fastMRI dataset and shows that MRI data varies significantly in appearance across different anatomies and image contrasts (FLAIR, T1, PD, etc). The image distribution also varies across vendors and magnetic field strengths of scanners, as the strength of the magnet impacts the signal-to-noise ratio (SNR), with stronger magnets leading to higher SNRs. The fastMRI dataset stands out for its diversity and size, making it particularly well-suited for exploring how different data distributions can affect the performance of deep learning models for accelerated MRI. We exploit this fact for our experiments in Section 3, 4, 5, and 6 by splitting the fastMRI dataset according to different attributes of the data. In Section 7, we showcase the generalizability of our findings by training models on all the datasets listed in Table 1, excluding the last four rows, which are reserved for robustness evaluation.

## 3 TRAINING A SINGLE MODEL OR SEPARATE MODELS ON DIFFERENT DISTRIBUTIONS

We start with studying whether training a model on data from a diverse set of distributions compromises the performance on the individual distributions. In its simplest instance, the question is whether a model for image reconstruction trained on data from both distributions $P$ and $Q$ performs as well on distributions $P$ and $Q$ as a model trained on $P$ and applied on $P$ and a model trained on $Q$ and applied on $Q$.

In general, this depends on the distributions $P$ and $Q$, and on the estimator. For example, consider a simple toy denoising problem, where the data from distribution $P$ is generated as $\mathbf{y} = \mathbf{x} + \mathbf{e}$, with $\mathbf{x}$ is drawn i.i.d. from the unit sphere of a subspace, and $\mathbf{e}$ is drawn from a zero-mean Gaussian with co-variance matrix $\sigma_P \mathbf{I}$. Data for distribution $Q$ is generated equally, but the noise is drawn from a zero-mean distribution with different noise variance, i.e., $\mathbf{e} \sim \mathcal{N}(0, \sigma_Q^2 \mathbf{I})$ with $\sigma_P^2 \neq \sigma_Q^2$. Then the optimal linear estimator learned from data drawn from both distribution $P$ and $Q$ is sub-optimal for both distributions $P$ and $Q$. However, there exists a non-linear estimator that is as good as the optimal linear estimator on distribution $P$ and distribution $Q$.

In addition, conventional approaches to MRI such as $\ell_1$-regularized least-squares need to be tuned individually on different distributions to achieve best performance, as discussed in Appendix B.

Thus it is unclear whether it is preferable to train a neural network for MRI on diverse data from many distributions or to train several networks and use them for each individual distribution. For example, is it better to train a network specific for knees and another one for brains or to train a single network on knees and brains together? Here, we find that training a network on several distributions simultaneously does not compromise performance on the individual distribution relative to training one model for each distribution.

**Experiments for training a joint or separate models.** We consider two distributions $P$ and $Q$, and train VarNets (Sriram et al., 2020), U-nets (Ronneberger et al., 2015), and ViTs (Lin & Heckel, 2022) on data $\mathcal{D}_P$ from distributions $P$ and on data $D_Q$ from distribution $Q$ separately. We also train a VarNet, U-net, and ViT on data from $P$ and $Q$, i.e., $\mathcal{D}_P \cup \mathcal{D}_Q$. We then evaluate on separate test sets from distribution $P$ and $Q$. We consider the VarNet because it is a state-of-the-art model for accelerated MRI, and consider the U-net and ViT as popular baseline models to demonstrate that our qualitative results are independent of the architecture. We consider the following choices of the datasets $\mathcal{D}_P$ and $\mathcal{D}_Q$, which are subsets of the fastMRI dataset specified in Figure 1:

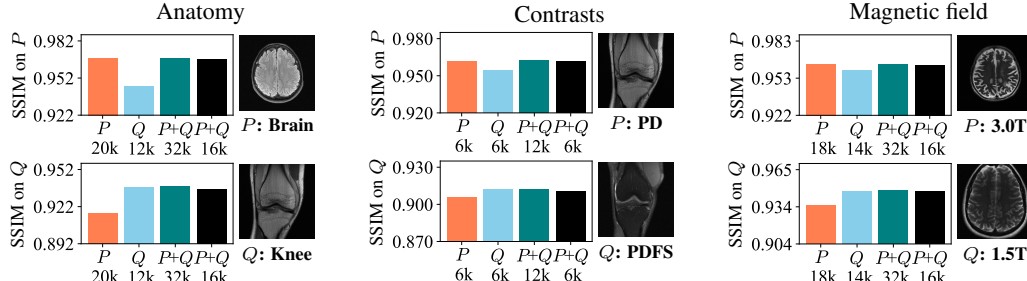

Figure 2: The **orange** and **blue** bars are the models (End-to-end VarNet) trained exclusively on data from $P$ ($\mathcal{D}_P$) and $Q$ ($\mathcal{D}_Q$), respectively, and the **teal** bars are the models trained on both sets $\mathcal{D}_P \cup \mathcal{D}_Q$. As a reference point, the **black** bars are the performance of models trained on random samples of $\mathcal{D}_P \cup \mathcal{D}_Q$ of **half the size**. The number below each bar is the total number of training images. It can be seen that we are in the high-data regime where increasing the dataset further gives minor improvements. For all distributions, the joint model trained on $P$ and $Q$ performs as well on $P$ and $Q$ as the models trained individually for each of those distributions.

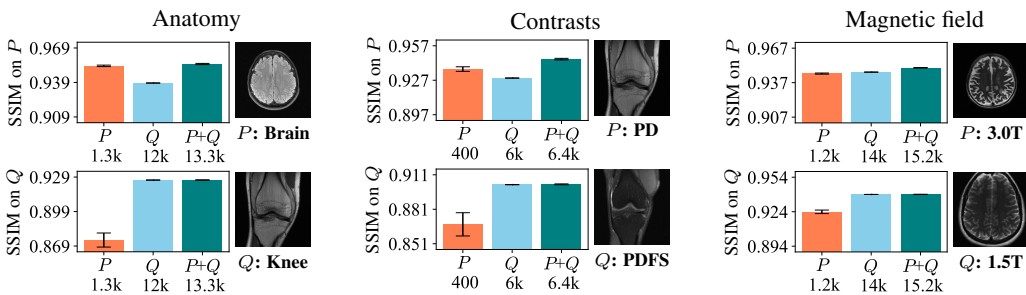

Figure 3: Training a single model on a skewed dataset does not harm performance on the individual data distributions. The number below each bar is the number of training examples used. We report the mean $\pm$ two standard deviations from five runs, each with a different random seed for sampling training data from $P$ and model initialization. We note for training sets exceeding 3k images, there is next to no variation (see Figure 14), therefore we only have error bars for this experiment which includes training runs on small datasets.

- **Anatomies.** $P$ are knees scans collected with 6 different combinations of image contrasts and scanners and $Q$ are the brain scans collected with 10 different combinations of image contrasts and scanners.

- **Contrasts.** We select $P$ as PD-weighted knee images from 3 different scanners and $Q$ are PDFS-weighted knee images from the same 3 scanners.

- **Magnetic field.** Here, we pick $P$ to contain all 3.0T scanners and $Q$ to contain all 1.5T scanners regardless of anatomy or image contrast.

The results in Figure 2 for VarNet show that the models trained on both $P$ and $Q$ achieve essentially the same performance on both $P$ and $Q$ as the individual models. The model trained on both $P + Q$ uses more examples than the model trained on $P$ and $Q$ individually. To rule out the possibility that the joint model is only as good as the individual models since it is trained on more examples, we also trained a model on $P + Q$ with half the number of examples (obtained by randomly subsampling). Again, the model performs essentially equally well as the other models. The results for Unet and ViT are qualitatively the same as the results in Figure 2 for VarNet (see App. E.1), and indicate that our findings are independent of the architecture used.

Thus, separating datasets into data from individual distributions and training individual models does not yield benefits, unlike for $\ell_1$-regularized least squares or the toy-subspace example.

**Experiments for training a joint or separate models on skewed data.** Next, we consider skewed data, i.e., the training set $\mathcal{D}_P$ is by a factor of about 10 smaller than the training set $\mathcal{D}_Q$. The choices for distributions $P$ and $Q$ are as in the previous experiment. The results in Figure 3 show that even for data skewed by a factor 10, the performance on distributions $P$ and $Q$ of models (here U-net) trained on both distributions is comparable to the models trained on the individual distributions.

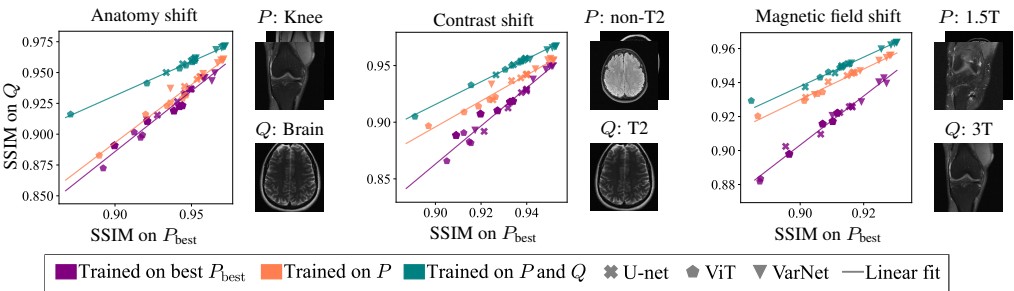

Figure 4: For a given distribution-shift from distributions $P = \{P_1, \ldots, P_m\}$ to distribution $Q$, we compare robustness of models trained on $P$ (**orange**) to baselines trained on the best single distribution $P_{\text{best}}$ (**violet**). As additional reference, we also report models trained on both $P$ and $Q$ to imitate ideally robust models (**teal**). Training on the more diverse dataset $P$ typically does not harm effective robustness compared to models trained on $P_{\text{best}}$, and is often even beneficial.

## 4 DATA DIVERSITY ENHANCES ROBUSTNESS TOWARDS DISTRIBUTION SHIFTS

We now study how training on diverse data affects the out-of-distribution performance of a model. We find that training a model on diverse data improves the models out-of-distribution performance.

**Measuring robustness.** Our goal is to measure the expected robustness gain by training models on diverse data, and we would like this measure to be independent of the model itself that we train on. We therefore measure robustness with the notion of *effective* robustness by Taori et al. (2020). We evaluate models on a standard 'in-distribution' test set (i.e., data from the same source that generated the training data) and on an out-of-distribution test set. We then plot the out-of-distribution performance of a variety of models as a function of the in-distribution performance, see Figure 4. It can be seen that the in- and out-of-distribution performance of models trained on data from one distribution, (e.g., in-distribution data **violet**) is well described by a linear fit. Thus, a dataset yields more effective robustness if models trained on it lie above the **violet** line, since such models have higher out-of-distribution performance for fixed in-distribution performance.

**Experiment.** We are given data from two distributions $P$ and $Q$, where distribution $P$ can be split up into $m$ sub-distributions $P_1, \ldots, P_m$. We consider the following choices for the two distributions, all based on the knee and brain fastMRI datasets illustrated in Figure 1:

- **Anatomy shift:** $P_1, \ldots, P_6$ is knee data collected with 6 different combinations of image contrasts and scanners, and $Q$ are the different brain datasets collected with 8 different combinations of image contrasts and scanners. To mitigate changes in the forward map (1) on this distribution shift, we excluded the brain data from the scanner Avanto as this data was collected with significantly fewer coils compared to the data from the knee distributions.
- **Contrast shift:** $P_1, \ldots, P_5$ are FLAIR, T1POST, or T1-weighted brain images and $Q$ are T2-weighted brain data.
- **Magnetic field shift:** $P_1, \ldots, P_7$ are brain and knee data collected with 1.5T scanners (Aera, Avanto) regardless of image contrast and $Q$ are brain and knee data collected with 3T scanners (Skyra, Prisma, Biograph mMR) regardless of image contrast.

For each of the distributions $P_1, \ldots, P_m$ we construct a training set with 2048 images and a test set with 128 images. We then train U-nets on each of the distributions $P_1, \ldots, P_m$ separately and select from these distributions the distribution $P_{\text{best}}$ that maximizes the performance of the U-net on a test set from the distribution $Q$.

Now, we train a variety of different model architectures including the U-net, End-to-End VarNet (Sriram et al., 2020), and vision transformers (ViT) for image reconstruction (Lin & Heckel, 2022) on data from the distribution $P_{\text{best}}$, data from the distribution $P$ (which contains $P_{\text{best}}$), and data from the distribution $P$ and $Q$. We also sample different models by early stopping and by decreasing the training set size by four. We plot the performance of the models evaluated on the distribution $Q$ as a function of their performance evaluated on the distribution $P_{\text{best}}$. Details are in Appendix C.1 and D.

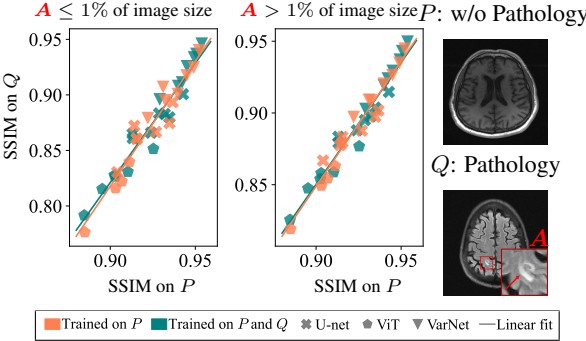

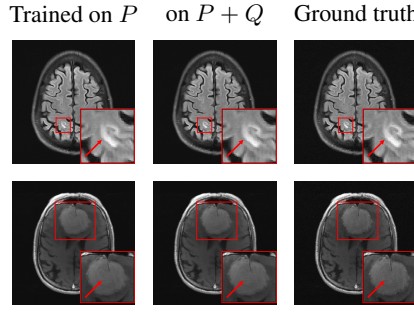

Figure 5: Models trained on images without pathologies can reconstruct pathologies as effectively as models trained on images with pathologies. SSIM is calculated only for the pathology region ($A$) for small pathologies (**left**) and for large pathologies (**right**).

Figure 6: Reconstructions given by the Var-Net of images containing a small-sized (**top**) or a large-sized (**bottom**) pathology, when trained on images without pathologies ($P$), and on images without and with pathologies ($P + Q$).

From the results in Figure 4 we see that the models **trained on $P$** are outperformed by the model **trained on $P$ and $Q$** when evaluated on $Q$, as expected, since a model **trained on $P$ and $Q$** is an ideal robust baseline (as it contains data from $Q$). The difference of the **trained on $P$ and $Q$**-line and the **trained on $P_{\text{best}}$**-line is a measure of the severity of the distribution shift, as it indicates the loss in performance when a model trained on $P_{\text{best}}$ is evaluated on $Q$. Comparing the difference between the line for the models **trained on $P$** and the line for models **trained on $P_{\text{best}}$** shows that out-of-distribution performance is improved by training on a diverse dataset, even when compared to the distribution $P_{\text{best}}$ which is the most beneficial distribution for performance on $Q$.

## 5 RECONSTRUCTION OF PATHOLOGIES USING DATA FROM HEALTHY SUBJECTS

In this section, we investigate the relevant distribution shift from healthy to non-healthy subjects. Specifically, we investigate how well models reconstruct images containing a pathology if no pathologies are contained in the training set. We find that models trained on fastMRI data without pathologies reconstruct fastMRI data with pathologies equally well as the same models trained on fastMRI data with pathologies.

**Experiment.** We rely on the fastMRI+ annotations (Zhao et al., 2022) to partition the fastMRI brain dataset into disjoints sets of images with and without pathologies. The fastMRI+ annotations (Zhao et al., 2022) are annotations of the fastMRI knee and brain datasets for different types of pathologies. We extract a set of volumes without pathologies by selecting all scans with the fastMRI+ label "Normal for age", and we select images with pathologies by taking all images with slice-level annotations of a pathology. The training set contains 4.5k images without pathologies ($P$) and 2.5k images with pathologies ($Q$). We train U-nets, ViTs, and VarNets on $P$ and on $P + Q$, and sample different models by varying the training set size by factors of 2, 4 and 8, and by early stopping. While the training set from distribution $P$ does not contain images with pathologies, $P$ is a diverse distribution containing data from different scanners and with different image contrasts.

Figure 5 shows the performance of each model evaluated on $Q$ as a function of its performance evaluated on $P$. Reconstructions are evaluated only on the region containing the pathology, where we further distinguish between small pathologies that make up less than $1\%$ of the total image size and large pathologies that make more than $1\%$ of the total image size. We see that the models **trained on $P$** show essentially the same performance (SSIM) as models **trained on $P + Q$** regardless of pathology size. The results indicate that models trained on images without pathologies can reconstruct pathologies as accurately as models trained on images with pathologies. This is further illustrated in Figure 6 (and Figure 17), where we show reconstructions given by the VarNet of images with a pathology: the model recovers the pathology well even though no pathologies are in the training set. Figure 18 in the appendix provides a more nuanced evaluation of the SSIM values for VarNet.

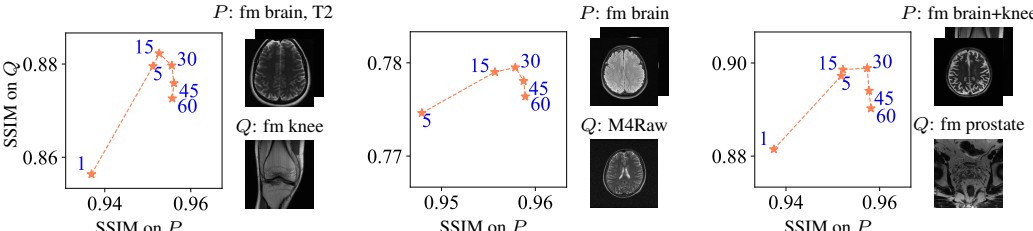

Figure 7: For a given distribution shift from distribution $P$ to distribution $Q$. The in- and out-of-distribution performance is plotted as a function of training epochs (1, 5, 15, 30, 45, 60). At the start of training, out-of-distribution performance increases together with in-distribution performance. Later on, out-of-distribution performance start to drop while in-distribution performance continues to increase marginally.

# 6    DISTRIBUTIONAL OVERFITTING

We observed that when training for long, while in-distribution performance continues to improve slightly, out-of-distribution performance can deteriorate. We refer to this as 'distributional overfitting'. Unlike 'conventional' overfitting, where a model's in-distribution performance declines after prolonged training, distributional overfitting involves a decline in out-of-distribution performance while in-distribution performance continues to improve (slightly). A similar observation has been made in the context of weight-space ensemble fine-tuning of CLIP (Wortsman et al., 2022).

Figure 7 illustrates distributional overfitting on three distribution shifts. Each plot depicts the in and out-of-distribution ($P$ and $Q$) performance of an U-net as a function of trained epochs (1, 5, 15, 30, 45, 60). For example, in the left plot $P$ is fastMRI T2-weighted brain data (fm brain, T2) and $Q$ is fastMRI knee data (fm knee). We observe as training progresses, the model initially improves both in-distribution and out-of-distribution performance. However, after epoch 15, out-of-distribution performance notably deteriorates, despite marginal improvements in in-distribution performance. This pattern is consistent across distribution shifts from the fastMRI brain/knee dataset to the M4Raw dataset and the fastMRI prostate dataset.

This finding indicates that early stopping, even before conventional overfitting sets in, can help to improve model robustness with minimal impact on in-distribution performance.

# 7    ROBUST MODELS FOR ACCELERATED MRI

The results from the previous sections based on the fastMRI dataset suggest that training a single model on a diverse dataset consisting of data from several data distributions is beneficial to out-of-distribution performance without sacrificing in-distribution performance on individual distributions. We now move beyond the fastMRI dataset and test whether this continues to hold on a large collection of datasets. We train a single large model for 4-fold accelerated 2D MRI reconstruction on the first 13 of the datasets in Table 1, which include the fastMRI brain and knee datasets, and use the remaining four datasets for out-of-distribution evaluation. The resulting model, when compared to models trained only on the fastMRI dataset, shows significant robustness improvements while maintaining its performance on the fastMRI dataset.

**Experiment.** We train an U-net, ViT, and the End-to-end VarNet on the first 13 datasets in Table 1. We denote this collection of datasets by $\mathcal{D}_P$. For the fastMRI knee and brain datasets, we exclude the official fastMRI knee validation set, as the fastMRI knee test set is not publicly available, and fastMRI brain test set from the training set. For the other datasets, we convert the data to follow the convention of the fastMRI dataset and omit slices that contain pure noise. The total number of training slices after the data preparation is 413k. For each model family, we also train a model on fastMRI knee, and one on fastMRI brain as baselines. To mitigate the risk of distributional overfitting, we early stop training when the improvement on the fastMRI knee dataset becomes marginal. We refer to Appendix C.2 and D for further details on the setup of this experiment.

We use the official fastMRI knee validation and fastMRI brain test set to measure in-distribution performance. We measure out-of-distribution performance on the last four datasets in Table 1, i.e., CC-359 sagittal view, Stanford 2D, M4Raw GRE, and NYU data. These datasets constitute a dis-

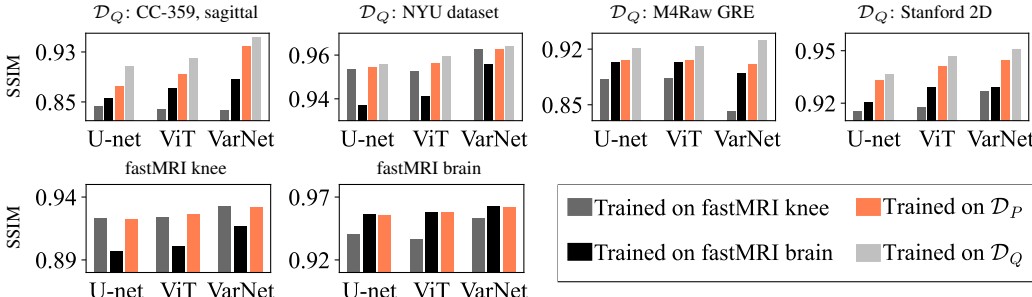

Figure 8: Training on a diverse collection of datasets improves robustness under distribution shifts. A model trained on the diverse set of datasets $\mathcal{D}_P$ can significantly outperform models trained on fastMRI data when evaluated on out-of-distribution data $\mathcal{D}_Q$, while maintaining the same performance on fastMRI data.

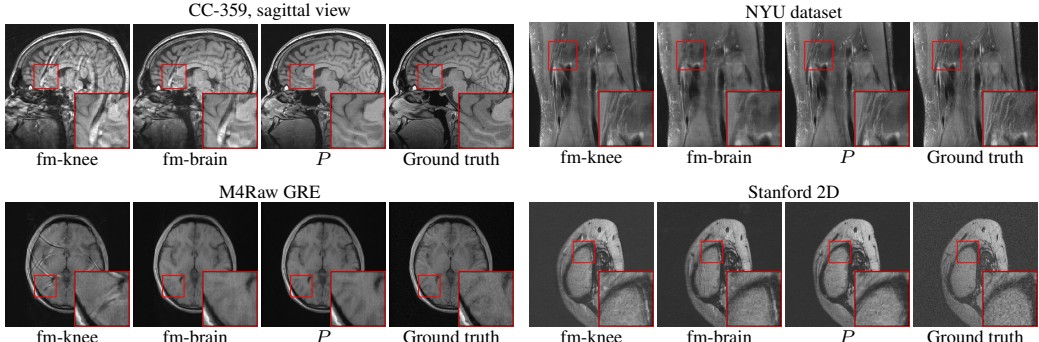

Figure 9: Random selection of reconstructions for the out-of-distribution datasets given by the VarNet (**top**) and U-net (**bottom**) trained on fastMRI knee (fm-knee), fastMRI brain (fm-brain), and collection of datasets $\mathcal{D}_P$. The model trained on $\mathcal{D}_P$ provides better details and fewer artifacts.

tribution shift relative to the training data with respect to vendors, anatomic views, anatomies, timeframe of data collection, anatomical views, MRI sequences, contrasts and combinations thereof and therefore enable a broad robustness evaluation. As a further reference point we also train models on the out-of-distribution datasets to quantify the robustness gap.

The results in Figure 8 show that for all architectures considered, the model trained on the collection of datasets, $\mathcal{D}_P$ significantly outperforms the models trained on fastMRI data when evaluated on out-of-distribution data, without compromising performance on fastMRI data. For example, on the CC-359 sagittal view dataset, the VarNet trained on $\mathcal{D}_P$ almost closes closes the distribution shift performance gap (i.e., the gap to the model trained on the out-of-distribution data). This robustness gain is illustrated in Figure 9 with examples images: the out-of-distribution images reconstructed by the model trained on the diverse dataset $\mathcal{D}_P$ have higher quality and fewer artifacts.

The results in this section reinforce our earlier findings, affirming that large and diverse MRI training sets can significantly enhance robustness without compromising in-distribution performance.

## 8    CONCLUSION AND LIMITATIONS

While our research shows that diverse training sets significantly enhance robustness for deep learning models for accelerated MRI, training a model on a diverse dataset often doesn't close the distribution shift performance gap, i.e., the gap between the model and the same idealized model trained on the out-of-distribution data (see Figure 4 and 8). Nevertheless, as datasets grow in size and diversity, training networks on larger and even more diverse data might progressively narrow the distribution shift performance gap. However, in practice it might be difficult or expensive to collect diverse and large datasets. Besides demonstrating the effect of diverse training data, our work shows that care must be taken when training models for long as this can yield to a less robust model due to distributional overfitting. This finding also emphasizes the importance of evaluating on out-of-distribution data.

## REPRODUCIBILITY STATEMENT

All datasets used in this work are referenced and publicly available. A comprehensive description of the data preparation process is provided in Appendix C, while details pertaining to model training can be found in Appendix D. For the implementation of our experiments, we heavily rely on the code provided by the fastMRI GitHub repository `https://github.com/facebookresearch/fastMRI/tree/main`. Our code will be made available to the public.

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

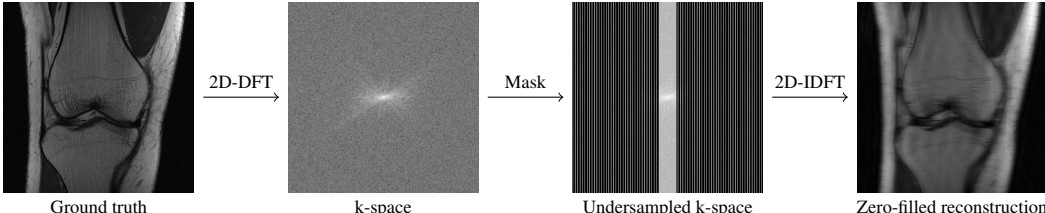

Figure 10: Illustration of the forward map for single-coil accelerated MRI. From left to right: The unknown ground-truth image first goes through a 2D-DFT. The Fourier spectrum (k-space) is then undersampled according to the acceleration factor by masking out frequency lines. Applying a 2D-IDFT to the masked k-space gives a coarse reconstruction which is often used as an input to a deep neural network.

Radhika Tibrewala, Tarun Dutt, Angela Tong, Luke Ginocchio, Mahesh B. Keerthivasan, Steven H. Baete, Sumit Chopra, Yvonne W. Lui, Daniel K. Sodickson, Hersh Chandarana, and Patricia M. Johnson. FastMRI Prostate: A Publicly Available, Biparametric MRI Dataset to Advance Machine Learning for Prostate Cancer Imaging, arXiv:2304.09254, 2023.

Mitchell Wortsman, Gabriel Ilharco, Jong Wook Kim, Mike Li, Simon Kornblith, Rebecca Roelofs, Raphael Gontijo Lopes, Hannaneh Hajishirzi, Ali Farhadi, Hongseok Namkoong, and Ludwig Schmidt. Robust fine-tuning of zero-shot models. In *IEEE/CVF Conference on Computer Vision and Pattern Recognition*, 2022.

Jure Zbontar, Florian Knoll, Anuroop Sriram, Tullie Murrell, Zhengnan Huang, Matthew J. Muckley, Aaron Defazio, Ruben Stern, Patricia Johnson, Mary Bruno, Marc Parente, Krzysztof J. Geras, Joe Katsnelson, Hersh Chandarana, Zizhao Zhang, Michal Drozdzal, Adriana Romero, Michael Rabbat, Pascal Vincent, Nafissa Yakubova, James Pinkerton, Duo Wang, Erich Owens, C. Lawrence Zitnick, Michael P. Recht, Daniel K. Sodickson, and Yvonne W. Lui. fastMRI: An Open Dataset and Benchmarks for Accelerated MRI, arXiv:1811.08839, 2019.

Ruiyang Zhao, Burhaneddin Yaman, Yuxin Zhang, Russell Stewart, Austin Dixon, Florian Knoll, Zhengnan Huang, Yvonne W. Lui, Michael S. Hansen, and Matthew P. Lungren. fastMRI+, Clinical pathology annotations for knee and brain fully sampled magnetic resonance imaging data. *Scientific Data*, 2022.

## A  ILLUSTRATION OF THE FORWARD MAP FOR ACCELERATED MRI

Figure 10 illustrates the forward map (1) for single-coil accelerated MRI, i.e., $C = 1$ and $S$ is identity. The unknown ground-truth image first goes through a 2D-DFT. The Fourier spectrum (k-space) is then undersampled according to the acceleration factor by masking out frequency lines. Applying a 2D-IDFT to the masked k-space gives a coarse reconstruction which is often used as an input to a deep neural network such as the U-net.

## B  $\ell_1$-REGULARIZED LEAST SQUARES REQUIRES DIFFERENT HYPERPARAMETERS ON DIFFERENT DISTRIBUTIONS

The standard non-deep learning approach for accelerated MRI is $\ell_1$-regularized least-squares (Lustig et al., 2007). While $\ell_1$-regularized least-squares is not considered data-driven, the regularization hyperparameter it typically chosen in a data-driven manner. For different distributions like different anatomies or contrasts, the regularization parameter takes on different values and thus the method needs to be tuned separately for different distributions. This can be seen for example from Table 4 of Zbontar et al. (2019).

To demonstrate this, we performed wavelet-based $\ell_1$-regularized least-squares on the single-coil knee version of the fastMRI dataset (Zbontar et al., 2019) using 100 images from distribution $P$: PD Knee Skyra, 3.0T and 100 from distribution $Q$: PDFS Knee Aera, 1.5T. Using a regularization weight $\lambda = 0.01$ on distribution $P$ gives a SSIM of 0.792, while $\lambda = 0.001$

yields subpar SSIM of 0.788. Contrary, on distribution $Q$, $\lambda = 0.01$ only yields 0.602, while $\lambda = 0.001$ yields SSIM 0.609. Thus, using the same model (i.e., the same regularization parameter for both distributions) is suboptimal for $\ell_1$-regularized. least squares. We used the BART `https://mrirecon.github.io/bart/` for the implementation of this experiment.

## C  DATA PREPARATION

### C.1  TRAINING SETS AND TEST SETS FOR EXPERIMENTS USING SUBSETS OF THE FASTMRI KNEE OR BRAIN DATASET

For the experiments in Sections 3, 4, 6, we construct training and test set as follows: For each of the distributions in Figure 1 we randomly sample volumes from the fastMRI training set for training and validation set for testing, such that the total number of slices amounts to around 2048 and 128, respectively. Training sets of combination of distributions are then constructed by aggregating the training data from the individual distributions. For example, if we consider distribution $P$ to contain all the 6 knee distributions from Figure 1 then the corresponding training set has $6\times2048$ training images. Likewise, if we consider $Q$ for example to contain all T2-weighted brain images the corresponding training set has $5\times2048$ training images. We note for Section 4, the models trained on $P$ and $Q$ are in fact trained on all the 16 distributions from Figure 1. This means that these models are always trained on $P$ and $Q$, regardless of how we choose $P$ and $Q$.

### C.2  PREPARATION OF DATASETS THAT ARE NOT FASTMRI KNEE OR FASTMRI BRAIN

We convert all the dataset listed in Table 1 to follow the fastMRI convention, where the anatomies in images are vertically flipped, targets are RSS reconstructions, and the k-space is oriented such that the horizontal axis corresponds to the phase-encoding direction and the vertical axis corresponds to the read-out direction.

If predefined train and test splits are not already provided with a dataset, we randomly select 85% of the volumes as training set and the remaining volumes as test set. For 3D MRI volumes, we synthesize 2D k-spaces by taking the 1D IFFT in the 3D k-space along either x, y or z dimension to create 2D volumes of different anatomical views (axial, sagittal and coronal). However, for the SKM-TEA dataset, we only consider the sagittal view. Depending on the dataset, the first and last 15-70 slices of the synthesized 2D volumes are omitted as we mostly observe pure noise measurements:

- CC-359, sagittal view: First 15 and last 15 slices are omitted.
- CC-359, axial view: First 50 slices are omitted.
- CC-359, coronal view: First 25 and 15 slices are omitted.
- Stanford 3D, axial view: First 5 and last 5 slices are omitted.
- Stanford 3D, coronal view: First 40 and last 40 slices are omitted.
- Stanford 3D, sagittal view: First 30 and last 30 slices are omitted.
- 7T database, axial view: First 70 and last 70 slices are omitted.
- 7T database, coronal view: First 30 and last 30 slices are omitted.
- 7T database, sagittal view: First 30 and last 30 slices are omitted.

For the other datasets that are not mentioned above, all slices are used. Moreover, each of the volumes of the SKM-TEA dataset originally contained two echos due to the qDESS sequence. We separated the two echos and counted them as separate volumes.

**fastMRI prostate T2.**  Originally, each volume of the fastMRI prostate T2 dataset contains three averages (Tibrewala et al., 2023): two averages sampling the odd k-space lines and one average sampling the even k-space lines. Then, for each average the authors estimate the missing k-space lines with GRAPPA (Griswold et al., 2002) and perform SENSE (Pruessmann et al., 1999) reconstruction. The final ground truth image is then obtained by taking the mean across the three averages (see code in author's GitHub Repository). However, we convert the data as follows: we take the raw k-space and average the two averages corresponding to the odd k-space lines and then fill the missing even

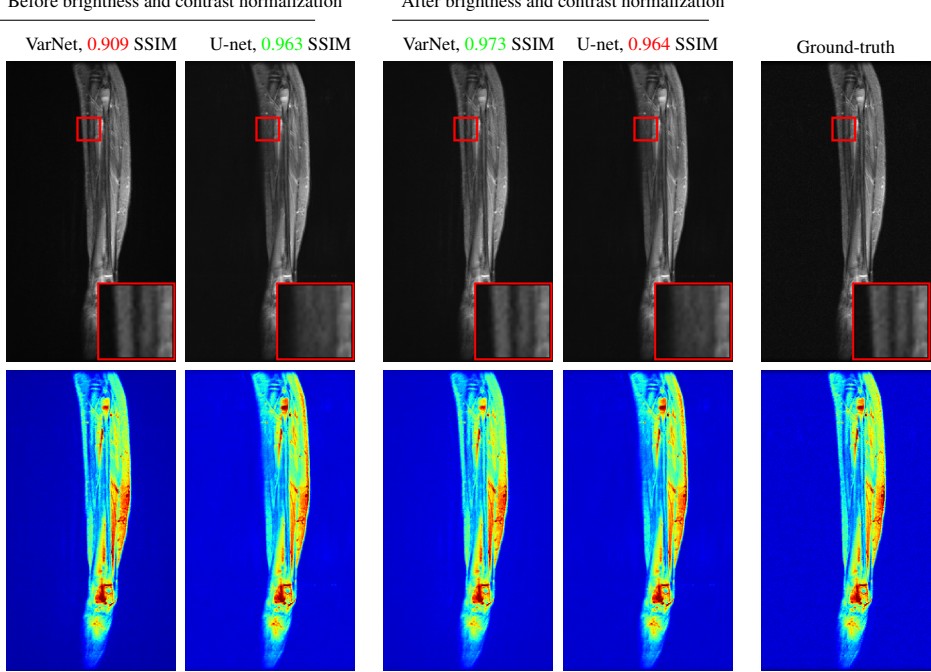

Before brightness and contrast normalization    After brightness and contrast normalization

VarNet, 0.909 SSIM    U-net, 0.963 SSIM    VarNet, 0.973 SSIM    U-net, 0.964 SSIM    Ground-truth

Figure 11: A subtle mismatch in terms of brightness and contrast between reconstruction and ground-truth can lead to a drastic loss in SSIM. Normalizing mean and variance of the reconstructions increases SSIM (without affecting the content) can result in a more faithful ranking of models. Top row and bottom row depict the same images, only differing in the color-map. All models were trained on the fastMRI knee dataset and applied to an example from the Stanford 2D dataset. Without normalization, the VarNet reconstruction has a slightly darker background compared to the other reconstructions (better seen in the bottom row). This leads to significantly lower SSIM, which is also lower than the U-net reconstruction, even though more details are recovered. After normalization, the VarNet obtains higher SSIM than the U-net which is consistent with the qualitative assessment of the reconstructions.

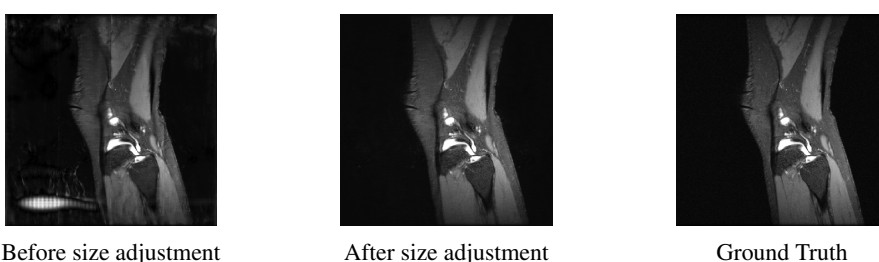

Before size adjustment          After size adjustment          Ground Truth

Figure 12: Mitigation of the artificial distribution shift due to mismatch in image size between training and test time. The End-to-end VarNet is trained on fastMRI knee dataset and applied to the Stanford 3D sagittal view dataset.

k-space lines with the average corresponding to the even k-space lines. This k-space serves as our raw k-space data. We then take this k-space data and apply a 2D-IFFT and finally perform a RSS reconstruction and use this image as ground truth.

## D  MODELS, TRAINING, AND EVALUATION

We consider U-nets with 4 pooling layers and 32 channels in the first pooling layer. The implementation of the model is taken from the fastMRI GitHub repository. Our configuration of the vision transformer (ViT) for image reconstruction is the same as 'ViT-S' from Lin & Heckel (2022), and the code is taken from the paper's GitHub repository. For the model input, we first fill missing

k-space values with zeros, then apply 2D-IFFT, followed by a root-sum-of-squares (RSS) reconstructions to combine all the coil images into one single image, and lastly normalize it to zero-mean and unit-variance. The mean and variance are added and multiplied back to the model output, respectively. This is a standard prepossessing step, as can be seen in the code of fastMRI's GitHub repository. The models are trained end-to-end with the objective to maximize SSIM between output and ground-truth.

Our configuration of the End-to-end VarNet (Sriram et al., 2020) contains 8 cascades, each containing an U-net with 4 pooling layers and 12 channels in the first pooling layer. The sensitivity-map U-net has 4 pooling layers and 9 channels in the first pooling later. The code for the model is taken from fastMRI's GitHub repository.

For any model and any choice of distributions $P$ or $Q$, the model is trained for a total of 60 epochs and use the Adam optimizer with $\beta_1 = 0.9$ and $\beta_2 = 0.999$. The mini-batch size is set to 1. We use linear learning-rate warm-up until a learning-rate of 1e-3 is reached and linearly decay the learning rate to 4e-5. The warm up period amounts to 1% of the total number of gradient steps. Gradients are clipped at a global $\ell_2$-norm of 1. During training, we randomly sample a different undersampling mask for each *mini-batch* independently. During evaluation, we generate for each *volume* the undersampling mask randomly which is then fixed and used for all slices within the volume.

The learning-rate for each model is tuned based on a grid search on the values $\{0.0013, 0.001, 0.0007, 0.0004\}$ and training on a random subset (2k slices) of the fastMRI dataset. We found negligible differences between learning rates 0.0013, 0.001, and 0.0007 when ensuring a sufficient number of epochs, and therefore keep the learning rate to 0.001 for simplicity. We also did the same grid search on fastMRI subsets for PD-weighted knee and PDFS-weighted knee and made the same observation.

In Section 7, the U-net has 124M parameter with 4 pooling layers and 128 channels in the first pooling layer. The maximal learning rate is set to 4e-4. The ViT has 127M parameters, where we used a patch-size of 10, an embedding dimension of 1024, 16 attention heads, and a depth of 10. The maximal learning rate is set to 2e-4. The VarNet is the same as described above. The learning-rate scheduler, with the same hyperparameters as mentioned earlier, is set for 40 epochs but we early stopped the models at epoch 24. The VarNet is trained for 40 epochs, as we use the same configuration as in the earlier experiments and have observed that training longer than 40 epochs gives marginal improvement. For U-net and ViT we use a mini-batch size of 8. For VarNet, we use a mini-batch size of 4. Since slice dimensions can vary across different volumes, the images within a mini-batch are chosen randomly from the same volume without replacement. Training was carried out on two NVIDIA RTX A6000 GPUs. Training the U-net took 384 GPU hours, the ViT took 480 GPU hours, and the VarNet took 960 GPU hours.

### D.1 OUT-OF-DISTRIBUTION EVALUATION

When evaluating our models from Section 7 on individual slices of the Stanford 2D dataset, we observed an anomaly: the VarNet occasionally yielded lower SSIM scores than the U-net, despite producing reconstructions of superior quality and accuracy. Upon investigation, we found that, under distribution shifts, the models sometimes struggled to precisely capture the brightness and contrast of an image. Even minor variations in these aspects can significantly impact the SSIM score. Given that radiologists routinely adjust the brightness and contrast of MRI images during inspection through a process known as 'windowing' (Ishida et al., 1984), we normalize the model output and target to have the same mean and variance during evaluation. This normalization reduces the SSIM score's sensitivity to variations in brightness and contrast, enabling it to better reflect structural differences, see Figure 11 for an illustration. Although Figure 11 primarily highlights the issue with VarNet, we note that U-net and ViT are also equally affected. We applied this normalization across all models in our evaluation in Section 7.

### D.2 RESOLUTION MISMATCH WITH VARNET ON OUT-OF-DISTRIBUTION EVALUATION

While the U-net and ViT are trained on *center-cropped* zero-filled reconstructions, the VarNet is trained on the entire k-space and therefore on the full-sized image, for example, the average im-

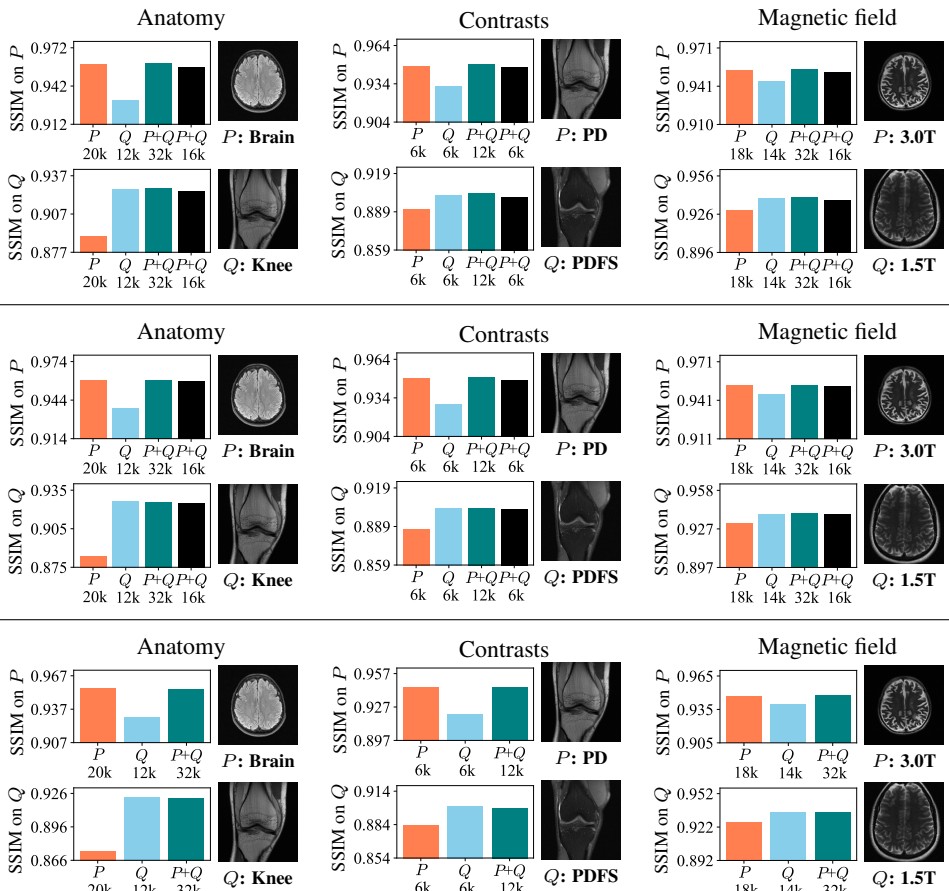

Figure 13: The **orange** and **blue** bars are the ViT (**Top**) and U-net (**Middle**) trained exclusively on data from $P$ ($\mathcal{D}_P$) and $Q$ ($\mathcal{D}_Q$), respectively, and the **teal** bars are the models trained on both sets $\mathcal{D}_P \cup \mathcal{D}_Q$. As a reference point, the **black** bars are the performance of models trained on random samples of $\mathcal{D}_P \cup \mathcal{D}_Q$ of **half the size**. The number below each bar is the total number of training images. It can be seen that we are in the high-data regime where increasing the dataset further gives minor improvements. For all distributions, the joint model trained on $P$ and $Q$ performs as well on $P$ and $Q$ as the models trained individually for each of those distributions. Similar results can observed when decreasing the size of the U-net by a factor of 10 (757k parameters, **Bottom**).

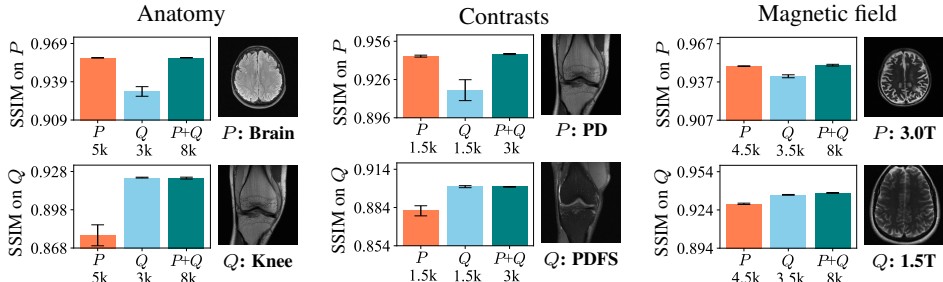

Figure 14: Also for smaller sized datasets, a single model is better than separate models. We report the mean $\pm$ two standard deviations from five runs with the U-net, each with a different random seed for sampling training data and model initialization. Note that when training on datasets with more than 3k images, there is next to no variation.

age size of fastMRI knee dataset is $640 \times 360$. Now, for example, when training the VarNet on the fastMRI knee dataset and evaluating it on the Stanford 3D dataset, which contains images of approximately half the size, we additionally introduce an 'artificial' distribution-shift by having a mismatch between image size during training and evaluation. To mitigate this artificial distribution-shift we

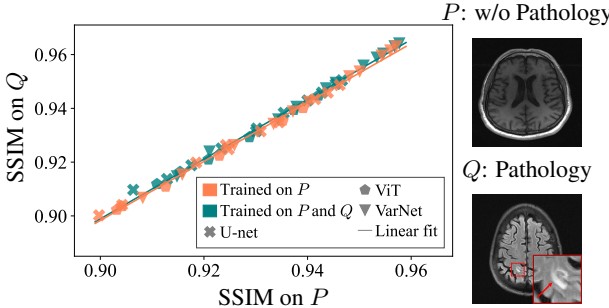

Figure 15: Models trained only on images without pathologies and models trained on images with pathologies have similar global SSIM. Different models are sampled by varying the training set size by factors of 2, 4 and 8, and by early stopping.

implement the following steps during inference: Given the undersampled k-space and mask, we first repeat the undersampled k-space one time in an interleaved fashion in horizontal and another time in vertical direction, and apply the same process to the undersampling mask. This input is fed to the VarNet and the output is center-cropped to the original image size. As can be seen in Figure 12, these processing steps heavily reduce artifacts. We apply this in Section 7 to the VarNets trained on the fastMRI datasets when evaluated on the Stanford 2D, CC-359 sagittal view, and M4Raw GRE dataset.

# E    ADDITIONAL RESULTS

## E.1    SINGLE MODEL VS. SEPARATE MODEL FOR OTHER MODEL TYPES

In Section 3, we show that training a single VarNet on two distribution gives the same performance a separate VarNet trained on the individual distributions. In Figure 13, we see that this result holds true also for the U-net and ViT. Figure 14 shows the same experiment for the U-net on smaller datasets.

## E.2    ADDITIONAL EVALUATIONS OF MODELS TRAINED ON HEALTHY SUBJECTS

Figure 18 presents the reconstruction performance evaluated for individual images in the test set, focusing on small pathologies. The evaluation specifically targeted the pathology regions. Results are provided for VarNet trained solely on images without pathologies ($P$) and VarNet trained on images with and without pathologies ($P + Q$). The results are presented for models where both models exhibited similar mean SSIM values for test images without pathology (approximately 0.957 SSIM) and also similar SSIM values for test images with small pathologies (approximately 0.948 SSIM).

While both models demonstrate high reconstruction performance for the majority of samples, indicated by high SSIM scores, a notable divergence can be observed in the low SSIM regime. In this regime, where certain samples are generally difficult to reconstruct (as even the model trained on $P + Q$ struggles), the variance between the two models increases. This means that for certain samples, the model trained on $P + Q$ outperforms significantly, whereas for other samples, the model trained solely on $P$ exhibits superior performance.

In Figure 17 we provide more reconstruction examples for small pathologies, obtained by the models from Section 5.

Moreover, in Section 5, we evaluate the model locally on the pathology region and observe models trained on image without pathologies perform the same as models trained on image with pathologies. In Figure 15, we show the results when SSIM is calculated across the entire image and not just for the pathology region.

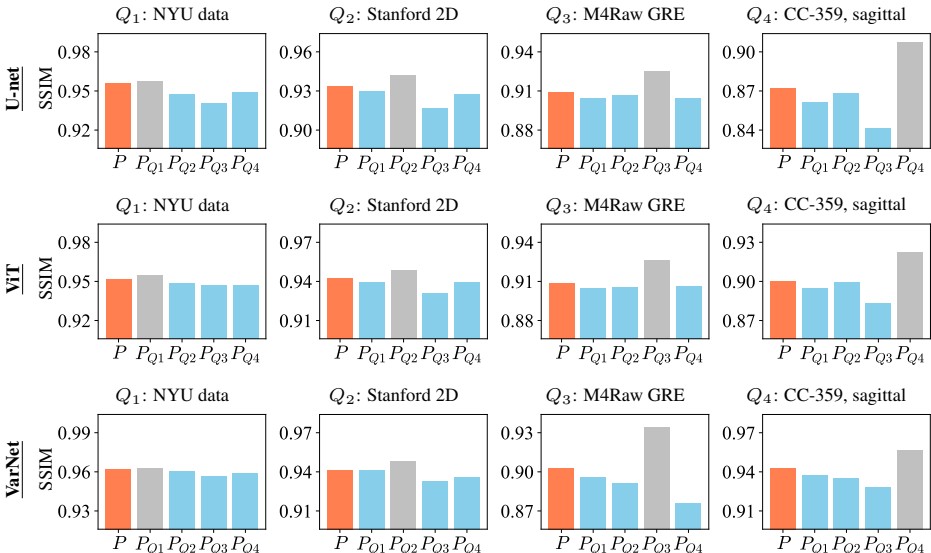

Figure 16: Fine-tuning deteriorates robustness. The fine-tuned model $P_{Q_i}$, obtained by fine-tuning the model trained on $\mathcal{D}_P$, which we denote by $P$, on *one* of the distributions $Q_1$, $Q_2$, $Q_3$ or $Q_4$, exhibits worse performance than model $P$ on out-of-distribution datasets.

## E.3   FINETUNING DETERIORATES ROBUSTNESS

Our results show that training a model on a diverse dataset enhances its robustness towards natural distribution-shifts. In this section we show that fine-tuning an already diversely trained model on a new dataset reduces its overall robustness. For this experiment, we take the models from Section 7 that were trained on $\mathcal{D}_P$ and fine-tune them on one of the four out-of-distribution datasets $\mathcal{D}_{Q_i}$. We denote the model fine-tuned on $Q_i$ by $P_{Q_i}$. As depicted in Figure 16, the fine-tuned model $P_{Q_i}$ exhibits improved performance on the specific data $Q_i$, as expected. However, it notably underperforms on all other datasets in comparison to the model trained on $\mathcal{D}_P$ (i.e., prior to fine-tuning).

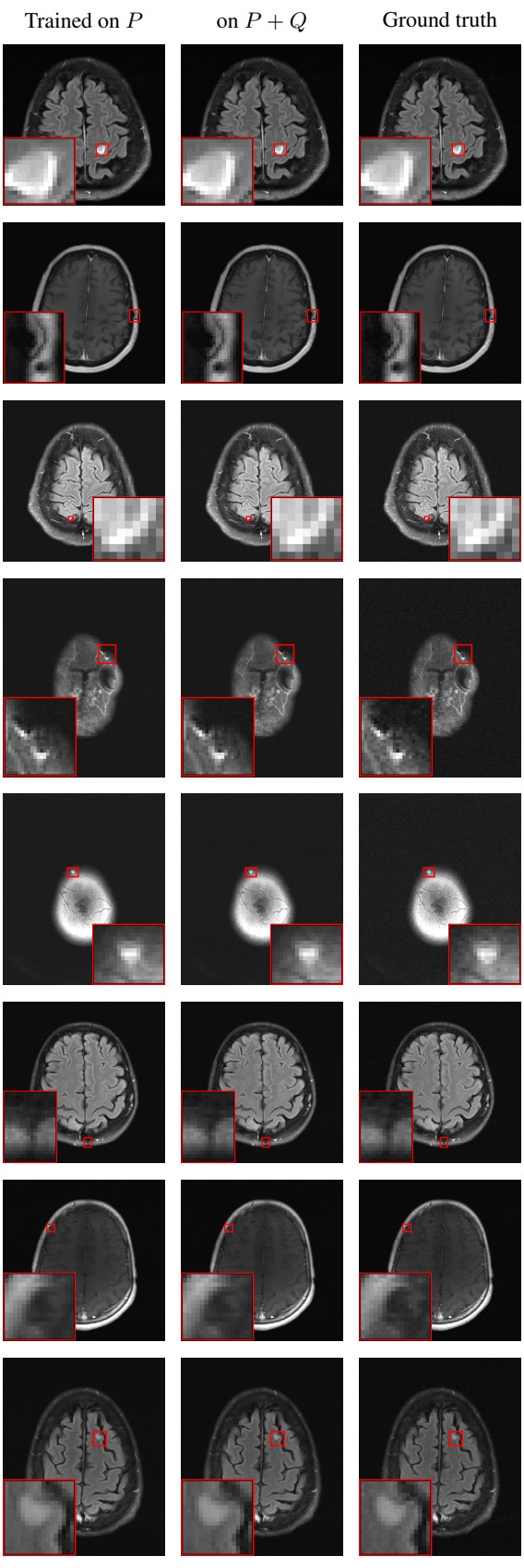

Figure 17: Random selection of reconstructions of *small* pathologies, given by the VarNet when trained on images without pathologies ($P$), and on images without and with pathologies ($P + Q$).

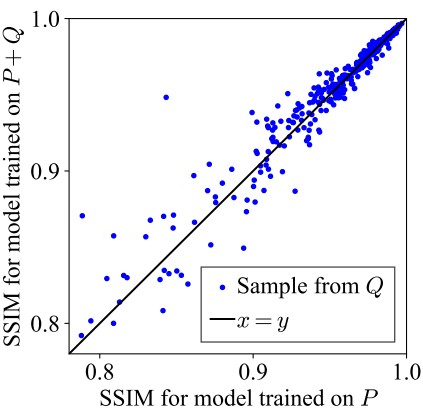

Figure 18: Reconstruction performance for each small pathology reconstructed with a VarNet trained only on data without pathologies (i.e., $P$) relative to the performance of a VarNet trained on data with and without pathologies ($P$ and $Q$). The SSIM is measured only within the region containing the pathology. Results are presented for the VarNet trained only on images without pathologies ($P$) and VarNet trained on images without and with pathologies ($P + Q$). The VarNet trained on $P$ and the VarNet trained on $P$ and $Q$ have very similar *mean* SSIM (0.957) on test images without pathology and similar SSIM (0.948) on test image with small pathologies. It can be seen that the majority of pathologies are reconstructed similarly well by both models (in terms of SSIM); however, in the regime where SSIM is low for either models (i.e., a regime where the patholgoies are inherently difficult to reconstruct) some images are reconstructed better by one model and others are reconstructed better by the other model.

