# OpenReview forum: "Robustness of Deep Learning for Accelerated MRI: Benefits of Diverse Training Data"
_ICLR.cc/2024/Conference — Submitted to ICLR 2024_

### Official Review · Reviewer_Eeae · 2023-10-27

**Soundness:** 2 fair
**Presentation:** 3 good
**Contribution:** 2 fair
**Rating:** 5
**Confidence:** 5

**Summary:**

This manuscript discusses the distributional robustness of deep learning based MRI reconstruction (solving an ill-posed inverse problem and recovering an underlying sub-Nyquist sampled image). The authors experimented with U-Net-based MRI reconstruction under multiple subtypes of distribution shifts and analyzed their effects on the performance. The authors also argue that more diverse data leads to more robust models.

**Strengths:**

The problem of identifying and mitigating distributional shifts for deep learning accelerated MRI is of significant real-world relevance. It is critical for building a trustworthy deep learning driven MRI reconstruction system.

The experiments are performed on a large range of MRI reconstruction datasets with multiple real-world types of distributional shifts (imbalanced data, anatomical shifts, diverse magnetic fields, images from healthy subjects or from patient with health conditions, etc.).

**Weaknesses:**

The scientific contributions of the manuscript is limited: despite the detailed analysis and discussion, the distributional robustness of deep learning MRI reconstruction has been discussed by a series of prior works [1-6]. Despite more detailed experiments (imbalanced data and healthy versus disease images), the major conclusions do not go beyond those of early works [1-3]. The authors failed to make significant theoretical and methodological contributions either (while most of [1-6] proposed either theoretical insights and/or methodological contributions).

The writing needs improvement: The paper is poorly structured, and it does not follow quite well the conventions of ICLR. It is difficult to identify the key arguments and contributions from the text. It is also difficult to grasp the chain of arguments and evidences.

Sec. 2: There is a lack of a brief introduction of essential key concepts: coils and sensitivity maps, sampling masks and accelerations, the signal-processing interpretation of MRI acquisition, problem settings for MRI reconstruction, etc. Missing these key concepts would bring difficulties for readers who are not familiar with MRI reconstruction.

The choice of using a U-Net is over-simplistic, given that the mainstream reconstructions works are based on unrolled proximal gradients with deep cascade networks, variational networks, as well as probabilistic diffusion models, which may also bring stronger distributional robustness due to better inductive biases compared with a plain U-Net.

[1] https://onlinelibrary.wiley.com/doi/10.1002/mrm.27355
[2] https://arxiv.org/abs/1902.10815
[3] https://onlinelibrary.wiley.com/doi/full/10.1002/mrm.28148
[4] https://arxiv.org/pdf/2011.00070.pdf
[5] https://link.springer.com/chapter/10.1007/978-3-030-87231-1_21
[6] https://www.sciencedirect.com/science/article/pii/S0730725X21002526

**Questions:**

The authors are encouraged to improve the clarity of the paper: talking about the problem background, existing works and their drawbacks, then the key contributions, in the introduction section. Then, in the following sections, the authors are encouraged to make their arguments clear, and then demonstrate how the experiments support their arguments.

The authors are also encouraged to bring more theoretical insight behind the observational results, given that distributional shifts in MRI are not a newly identified problem.

The authors are also encouraged to take the effect of different reconstruction technique into consideration: despite diverse implementation details, these methods can be generally categorized as 1. plain feed-forward networks; 2. unrolled cascaded networks; 3. variational networks, as well as 4. probabilistic diffusion models. The authors may want to consider the effects of inductive biases on distributional robustness.

**Details Of Ethics Concerns:**

This is a retrospective study based on public datasets. The authors need to adhere to terms and conditions specified by their owners.

---

> ### Author Response · Authors · 2023-11-19
>
> Thanks for noting that the problem we consider has 'significant real-world relevance' and that 'the experiments are performed on a large range of MRI reconstruction datasets with multiple real-world types of distributional shifts'.
>
> The main concern of the reviewer is whether our results go beyond those in [1-6]. Here is how our contribution are different from [1-6], and how they go significantly beyond those existing works:
>
> - F. Knoll et al. [1] and J. Huang et al. [6] characterise the severity of specific distribution-shifts, and suggest the use of transfer learning to avoid distribution shifts. Contrary, we investigate the role of training data diversity in influencing robustness and find that diverse training data can significantly improve robustness.
>
> - C. Ouyang et al. [2] propose an approach that modifies natural images for training MRI reconstruction models. Contrary, we consider models trained in the conventional way, i.e., training on raw k-space data, and analyse systematically how combining different data-distributions affects robustness, and we find that it can significantly improve out-of-distribution robustness.
>
> - Similar to F. Knoll et al. [1] and C. Ouyang et al. [2], SUH Dar et al. [3] propose transfer-learning to train a reconstruction model on natural images and then fine-tune it on MR images. The goal of their work is to increase training data efficiency. X. Liu et al. [5] propose a special network architecture to improve the performance of training on multiple anatomies simultaneously for. Their results are based on retrospectively under-sampled DICOM images and the method only works for emulated single-coil data. Both works focus on improving in-distribution performance and do not study the robustness issue. Our objective is to study how performance and robustness of models is affected by data diversity of raw k-space data.
>
> - F. Caliva et al. [4] consider the problem of adversarial robustness, whereas we consider the problem of robustness under \emph{natural} distribution-shifts.
>
> Additionally, the reviewer is concerned that the choice of studying U-net is over-simplistic as better models exist. We choose the Unet model since it is a well known baseline and our results are agnostic to the architecture. However, we agree it is important to explicitly show this, and have carried out a significant number of additional simulations after submitting the paper. Specifically, we have expanded our results in Section 3 and 7 with results obtained from the End-to-end VarNet (mentioned by the reviewer), a state-of-the-art model for MRI reconstruction, and also with results from ViT to demonstrate that our results continue to hold true even for non-convolutional architectures.
>
> The reviewer also raises ethics concerns regarding the datasets used in our work. We have checked the terms and conditions of each dataset carefully, and confirm that they are safe to use for research, and that no conditions are violated. Most of the dataset are specifically released for image reconstruction research, which is what we do in this paper.
> We hope that our response clarified the reviewer's concern and if yes would be happy if the reviewer would consider raising their score.
>
> References:
>
> [1] F. Knoll et al. “Assessment of the generalization of learned image reconstruction and the potential for transfer learning.” Magnetic Resonance in Medicine, 2019.
>
> [2] C. Ouyang et al. “Generalising Deep Learning MRI Reconstruction Across Different Domains.” arXiv:1902.10815, 2023.
>
> [3] SUH Dar et al. “A Transfer-Learning Approach for Accelerated MRI Using Deep Neural Networks.” Magnetic Resonance in Medicine, 2020.
>
> [4] F. Caliva et al. “Adversarial Robust Training of Deep Learning MRI Reconstruction Models.” arXiv:2011.00070, 2021.
>
> [5] X. Liu et al. “Universal Undersampled MRI Reconstruction.” Medical Image Computing and Computer Assisted Intervention, 2021.
>
> [6] J. Huang et al. “Evaluation on the generalization of a learned convolutional neural network for MRI reconstruction.” Magnetic Resonance in Medicine, 2022.
>
> [7] R. Taori et al. “Measuring Robustness to Natural Distribution Shifts in Image Classification.” In Advances in Neural Information Processing Systems, 2020.

---

> ### Comment · Reviewer_Eeae · 2023-11-21
> **Interesting work with practical value, but limited novelty/insight issue remains**
>
> I would like to thank the authors for the clarification, and I am pleased to see that the authors have experimented with Variational networks and ViT. These results would consolidate the conclusions in the manuscript.
>
> I would also like to thank the authors for clarifying their conclusions: 1. similar to those with image classification, deep networks also suffer from distributional shifts (in the manuscript, termed as *robustness*, although for most of readers of ICLR, the term *robustness* more often refers to adversarial robustness which the manuscript fails to discuss); 2. diversity of training data enhances distributional robustness. I am also pleased to see the authors to have read through the suggested existing works. With that being said, I would still kindly remind that these conclusions/contributions still may not go beyond those by [1-6]:
>
> 1. Deep MRI reconstructions suffer from robustness issue: This has been the main focus of [1,2,6] (robustness against distributional shifts) and [4] (robustness against adversarial perturbations). I appreciated the increased scales of datasets and experiments in the manuscript. However, readers may argue there were no significant breakthroughs over those by [1,2,4,6].
>
> 2. Incorporating training datasets with more diversities enhances the robustness: I agree this is an interesting empirical result. However, this conclusion is neither novel, nor rigorous:
>
> a) Novelty: The benefit of diversity of training data has been the focus of [2,5]. [2] finds that training with MS-COCO, which bears more diversity (at the patch scale) than application-specific medical datasets, yields the best overall distributional robustness (evaluating the model on datasets that are different from those for training). [5] finds that aggregating multiple different training datasets leads to the best overall performance compared with dataset-specific training (with their new workflow). Despite the increased scale of experiments in the manuscript, it is difficult to identify major breakthroughs beyond those reached by [2,5].
>
> b) Rigor: The term *diversity* is quite a sloppy umbrella term. Theoretically, the data diversity would only make sense when the diverse training data could better cover the distribution of the test data (at least better covers its support). However, this has been ignored by the authors, and adding further in-depth analysis/experiment requires substantial amount work that falls beyond the scope of a revision. Also, [5] finds that *... simply mixing all the datasets does not lead to an ideal model...* (Table 1, *Independent* versus *Shared*), which contradicts a bit to the claims of your manuscript. This result in [5] suggests that diverse training data does not necessarily lead to benefits, while the authors did not provide an in-depth analysis to the theoretical ground behind the scene.
>
> In summary, while the manuscript presents a practical problem setting in together with extensive empirical results, the relatively modest level of novelty and theoretical depth may not align well with the standards and scope of ICLR.

---

> > ### Author Response · Authors · 2023-11-22
> >
> > Thanks for the comment. We are glad that the reviewer finds the new experiments with VarNet and ViT to be valuable, the results to be practically useful, and our finding that diversity enhances robustness to be interesting.
> >
> > Thanks for the references [2,3,5,6] they are important related works we thus added them to the related work section of our paper ([1] was already referenced in the original version).
> >
> > However, we believe that our work provides important new results relative to the mentioned references as explained in our previous reply. In particular, none of the mentioned works has demonstrated that diversity improves robustness.
> >
> > Regarding rigor: We agree that diversity is a sloppy term; however, unlike all the works referenced by the reviewer, we study out-of-distribution robustness, and we do so rigorously with the \emph{effective robustness} framework [7], a recognized framework for studying robustness.
> >
> > In the results prior to our work, training on diverse datasets might have lead to a better model, but not necessarily more \emph{robust} models: Since in- and out-of-distribution performance is correlated for the distribution shifts we consider (see [11] and Figure 4 in our paper), training on some dataset can improve both in and out-of-distribution performance (it moves on the line) but not yield a more robust model beyond what is expected from the in-distribution performance. In our work we show for the first time in the context of accelerated MRI that training on more diverse data, leads to an \emph{effective} robustness gain in that it gives better out-of-distribution performance relative to the expected in-distribution performance.
> >
> > The Reviewer noted that our work does not go beyond [2, 5]. We respectfully disagree:
> >
> > - As mentioned in the previous reply, [2] proposes an approach that modifies natural images for training MRI reconstruction models, and does so on retrospectively undersampled magnitude images which is problematic, see [12]. We study on real-world k-space data how diversity affects real-world distribution shifts.
> >
> > - [5] proposes an approach to improve \emph{in-distribution} performance and \emph{does not} study out-of-distribution robustness. In addition, the study [5] is based on retrospectively undersampled magnitude images which is problematic, see [12].
> >
> > In addition, the reviewer states that our finding that training two models on two distributions separately is not more beneficial than training a single model on both distributions contradicts the claim by [5]: “simply mixing all the datasets does not lead to an ideal model” (see Table 1 in [5]). There is no contradiction, as [5] doesn't make a statement about out-of-distribution data, and we don't claim that training on different distributions leads to an ideal model for \emph{in-distribution} performance. Figure 2 shows comparable SSIM for \emph{in-distribution} performance between training two separate models and training a single model on both distributions. This aligns with the findings in Table 1 of [5], where the differences in PSNR and SSIM are marginal, reinforcing our statement.
> >
> > Regarding [4], this paper is about adversarial robustness [4], while we study robustness to distribution shifts. Those are different notions of robustness.
> >
> > We hope that this addresses the remaining concerns.
> >
> > References:
> >
> > [1] F. Knoll et al. “Assessment of the generalization of learned image reconstruction and the potential for transfer learning.” Magnetic Resonance in Medicine, 2019.
> >
> > [2] C. Ouyang et al. “Generalising Deep Learning MRI Reconstruction Across Different Domains.” arXiv:1902.10815, 2023.
> >
> > [3] SUH Dar et al. “A Transfer-Learning Approach for Accelerated MRI Using Deep Neural Networks.” Magnetic Resonance in Medicine, 2020.
> >
> > [4] F. Caliva et al. “Adversarial Robust Training of Deep Learning MRI Reconstruction Models.” arXiv:2011.00070, 2021.
> >
> > [5] X. Liu et al. “Universal Undersampled MRI Reconstruction.” MICCAI, 2021.
> >
> > [6] J. Huang et al. “Evaluation on the generalization of a learned convolutional neural network for MRI reconstruction.” Magnetic Resonance in Medicine, 2022.
> >
> > [7] R. Taori et al. “Measuring Robustness to Natural Distribution Shifts in Image Classification.” NeurIPS, 2020.
> >
> > [8] B. Recht et al. “Do ImageNet Classifiers Generalize to ImageNet?” ICML, 2019.
> >
> > [9] T. Nguyen et al. “Quality Not Quantity: On the Interaction between Dataset Design and Robustness of CLIP.” NeurIPS, 2022.
> >
> > [10] J. Miller et al. “Accuracy on the Line: On the Strong Correlation Between Out-of-Distribution and In-Distribution Generalization”. ICML, 2021
> >
> > [11] M. Darestani et al. “Measuring Robustness in Deep Learning Based Compressive Sensing.” ICML, 2021.
> >
> > [12] E. Shimron et al. “Implicit data crimes: Machine learning bias arising from misuse of public data”. PNAS, 2022.

---

### Official Review · Reviewer_JsA6 · 2023-10-29

**Soundness:** 2 fair
**Presentation:** 3 good
**Contribution:** 2 fair
**Rating:** 5
**Confidence:** 5

**Summary:**

The paper examines the effect of diverse training data on the performance of MRI reconstruction models. To perform this experiment, the paper considers a wide suite of datasets with fully-sampled raw data, including fastMRI, Stanford 3D, the 7T database, CC-359, and others. For most of these datasets, a U-Net is trained and then evaluated on data that would be considered out-of-domain from the training distribution. The paper has a series of conclusions based on empirical evidence:

1. Having separate models for each distribution is not better than having one model for all distributions. This includes skewed data situations.
2. Data diversity improves robustness to distribution shift.
3. Pathology can be reconstructed from healthy subjects.
4. Hold-out-sets with out-of-domain data can be used to assess overfitting.
5. A model trained on all data is most robust.

**Strengths:**

- This paper is one of the largest in terms of experiments on the effects of data for MRI reconstruction that I have seen so far.
- The experiments consider varying classes of models beyond the U-Net used for most experiments.
- The goals of the paper are clearly presented and examined in targeted experiments.
- The findings on out-of-distribution hold-out performance as a surrogate for early stopping could be useful to practitioners.

**Weaknesses:**

I am currently learning towards rejection because there are some issues with several of the conclusions.

1. For the first conclusion, no statistical tests or confidence intervals are used to qualify the statement that P+Q gives similar performance to P alone. I also think that a more rigorous analysis should have been done on a case level to compare the methods, as average SSIM scores can obscure what is going on at the tails. Medicine is inherently risk averse, so the tails are critical for this application.
2. For data diversity, this effect seems to somewhat rehash the results of (Knoll, 2019), but with larger quantities of data. As with (1), a deeper analysis of edge cases could have been useful. However, in general I don't have major issues with this section.
3. The analysis of pathology could be particularly problematic, as SSIM average scores obscure what is going on in small regions of the image. For example, in (Radmanesh et al., 2022) the SSIM for T1 images at a 6X acceleration is still quite high, but only 80% of the cases were accepted by radiologists for being clinically acceptable. In the same paper, Figure 1 shows that for very low SSIMs, some large and prominent pathologies can still be seen, whereas more modest pathologies such as MS lesions are erased at low accelerations. All this is to say that the SSIM metric is not indicative of performance for pathology cases, and a deeper analysis is needed to substantiate the claim that models trained on healthy subjects can reconstruct pathology.

So in summary, the paper's analysis is lacking in a few areas that probably need to be improved, including most importantly a deeper statistical analysis and looking at pathology beyond global SSIM numbers. Beyond that, the paper is submitted to the datasets and benchmarks track, but I did not identify what it considered to be its core contribution to that area.

Radmanesh, Alireza, et al. "Exploring the Acceleration Limits of Deep Learning Variational Network–based Two-dimensional Brain MRI." Radiology: Artificial Intelligence 4.6 (2022): e210313.

**Questions:**

1. Why did you elect to use the U-Net for most experiments rather than a VarNet, which is generally more popular in the field?
2. Could you provide more detail on the 7T database? The 7T database paper does not say much about a data release, and it only mentions 24 volunteers. Noting this, it might be good to list the number of subjects for each dataset in Table 1.
3. Did you consider transfer learning as an alternative path to gaining the benefits of diverse training, as originally proposed in (Knoll, 2019)? Using full data from the beginning could take substantially more compute and would be a limitation.
4. Did you consider the effect of model size on robustness?
5. The current paper considers discriminative models, but more many approaches have been proposed based on generative priors that might be robust and can be trained on non-fully-sampled data (e.g., Jalal, 2021). Did you consider looking at these generative models?

Jalal, Ajil, et al. "Robust compressed sensing mri with deep generative priors." Advances in Neural Information Processing Systems 34 (2021): 14938-14954.

---

> ### Author Response · Authors · 2023-11-19
>
> Many thanks for the review and for noting that this work is 'one of the largest in terms of experiments on the effects of data for MRI reconstruction', and that the findings can be useful for practitioners.
>
> - The reviewer raises concerns about (i) the lack of confidence intervals in Section 3, where we show that training a model on two distributions gives similar performance to training models on the distributions separately, and (ii) that global SSIM scores for evaluating pathology reconstructions (Section 5), could be problematic due to small pathology sizes.
>
> - Regarding concern (i), we changed Figure 3 to include the mean and standard deviations over 5 independent runs and added Figure 14 which also reports the mean and standard deviations over 5 independent runs for the same experiment as in Figure 2 but on smaller datasets. It can be seen from Figure 14 that the confidence intervals are extremely small (almost not visible) for models trained on more than 3k images, which concerns the vast majority of experiments in our paper, and thus we did not add confidence intervals throughout, since this wouldn't be justified given the cost of thousands of GPU hours and the fact that the confidence intervals would be very small.
>
> - As a response to concern (ii), we have now re-evaluated the models only on the region containing the pathology (instead of computing metrics on the entire image), see the revised Figure 5. We further distinguished between small pathologies and relatively large pathologies.
>
> - We also note that we do not claim that models trained only on healthy images can reconstruct pathologies. Rather, our results indicate that models trained on data without pathologies can reconstruct pathologies as accurately as models trained on data with pathologies.
> Regarding our reason to submit to the 'datasets and benchmark category': One category mentioned regarding the scope of the datasets and benchmarks tracks is often (e.g., at NeurIPS) 'Data-centric AI methods and tools, e.g. studies in data-centric AI that bring important new insight', we believe our work falls into this category.
>
> In the following we address the questions from the reviewer:
>
> - Answer to Q1:
> We have expanded our results in Section 3 and 7 with results obtained from the End-to-End VarNet and also with results from ViT to showcase that our results hold true even for non-convolutional architectures. We initially chose the U-net since we expect our results to be relatively agnostic to the architecture of the end-to-end network, and our results indicate that they are, and the U-net is easy and fast to experiment with.
>
> - Answer to Q2:
> The 7T database paper does not say much about a data release, and it only mentions 24 volunteers. Noting this, it might be good to list the number of subjects for each dataset in Table 1.
> We made a mistake with the citation and have now corrected it. We thank the reviewer for noticing this error. We also included the number of subjects in Table 1.
>
> - Answer to Q3:
> While transfer learning can definitely boost the performance on the target distribution, we do not advocate for it as it deteriorates robustness (see the new Appendix E.3 in the revised manuscript).
>
> - Answer to Q4:
> In Section 7, the models are substantially larger (except for VarNet) than in the previous sections and the results show that the conclusions from the previous sections still apply (i.e., diverse training sets give the same in-distribution performance but better out-of-distribution performance; hence, increased effective robustness).
>
> - Answer to Q5:
> While generative models offer an interesting path for investigation, we  focus exclusively on models trained end-to-end, since those models (in particular the end-to-end VarNet) are widely recognized in the community and currently achieve state-of-the-art results in accelerated MRI.
>
> If our responses and the new experiments and changes in the paper have alleviated the reviewers' concerns we would be happy if the reviewer would consider raising their score.
>
> References:
>
> F. Knoll et al. “Assessment of the generalization of learned image reconstruction and the potential
> for transfer learning.” Magnetic Resonance in Medicine, 2019.
>
> R. Taori et al. “Measuring Robustness to Natural Distribution Shifts in Image Classification.” In Advances in Neural Information Processing Systems, 2020.
>
> T. Nguyen et al. “Quality Not Quantity: On the Interaction between Dataset Design and Robustness of CLIP.” In Advances in Neural Information Processing Systems, 2022.

---

> > ### Comment · Reviewer_JsA6 · 2023-11-20
> >
> > First I want to thank the authors for putting so much work into updating their paper. However, I still have some outstanding concerns for both (i) and (ii).
> >
> > (i) Unfortunately, I think there was a misunderstanding. Typically, variance over model runs is not the primary source of variance. Rather, it is variance over patients. So I was expecting confidence intervals on a patient level and whether the mean is shifted between the two distributions beyond the noise level related to the SSIM scores in a patient population. This would ideally be only the first step of statistical analysis that would be followed up by looking at edge cases and tails.
> >
> > (ii) For this, I do not think that evaluating pathology on only the pathology region is sufficient, since SSIM fundamentally measures a different task than pathology detection. Ideally, this would be done in conjunction with review of the images by clinical radiologists and using a biostatistics protocol more common to that field. So my assessment is the claims of the pathology reconstruction section remain over-broad, or at least are not presented in a properly sensitive-enough manner.
> >
> > Since these were the primary two aspects for my original review, I have to maintain my original score.

---

> > > ### Author Response · Authors · 2023-11-21
> > >
> > > Thank you for answering and for reading the revised paper.
> > >
> > > Regarding point (i), we have already considered the variance over model runs and over patients. See the caption of Figures 3 and 14, where we say that different training data (i.e., patients in our case) are sampled for each run. Indeed, the variance from the patients is larger than from the random initialization, but that is already reflected in the confidence intervals. We hope this clarification addresses the Reviewer's concern.
> > >
> > > Regarding point (ii), we would like to emphasize that we do not make any statements about the broader implications of the reconstructions, such as pathology detection. We only state, based on our results in Section 5, that models trained without pathology images can reconstruct a pathology as well as models trained on data including pathology images (regardless of how accurate the reconstructions are relative to the ground truth).
> > >
> > > To provide additional evidence, we have added Figure 17 in the Appendix showing pathologies reconstructions with (A) a model trained on data without pathologies and (B) a model trained on data with and without pathologies. It can clearly be seen that the reconstructions of the pathologies provided by the models A and B are indistinguishable, as the SSIM values have already indicated (see Figure 5).
> > >
> > > We also talked with a radiologist about our finding, and the radiologist mentioned that there is nothing to evaluate here, since the reconstructions are indistinguishable. Recall again that we only make a statement about the difference in the reconstruction performance between the models A and B, not about broader statements on the reconstructions of pathologies.

---

> > > > ### Comment · Reviewer_JsA6 · 2023-11-21
> > > >
> > > > For (i), I was expecting intervals as large as 0.07, as reported in the 2020 fastMRI challenge paper. This would be larger than the small intervals currently reported. From the text for Figure 3, it sounds like five training runs were used to give five mean SSIM samples, and a confidence interval was calculated over this sample of five. The confidence intervals that would be more informative would be one over a single realization of the test set around the test set mean SSIM. This would have required no extra compute. Similarly, with two treatments, you could have done a paired t-test on the SSIMs for each sample and likely concluded a difference. Note that neither of these include a tail analysis, which was one of my other original concerns.
> > > >
> > > > For (ii), the last paragraph of Section 5 (along with any other related sections of the paper) would have to be altered to limit the conclusions to SSIM accuracy and SSIM accuracy only, and that further validation for pathology detection would be necessary to assess clinical feasibility.
> > > >
> > > > I am unable to evaluate the images. A radiologist study is difficult to do. Some images are easier to reconstruct than others, so the evaluation set has to be carefully chosen. It would be easier to alter the conclusions of the paper and mention limitations.

---

> > > > > ### Author Response · Authors · 2023-11-22
> > > > >
> > > > > Many thanks for discussing this important point with us, we really appreciated it, and we feel we now understand where the misunderstanding lies and would like to clarify.
> > > > >
> > > > > Regarding (ii): We revised section 5 on pathologies further and added Figure 18 to the paper with an additional nuanced result on the reconstruction of the pathologies, including analysis of the tails as requested. The core claim we make in the revised section is "We find that models trained on fastMRI data without pathologies [model A] reconstruct fastMRI data with pathologies equally well [model B] than the same models trained on fastMRI data with pathologies." We now specify that this claim pertains to the setup we consider, i.e., the FastMRI training set along with the annotations, and for this claim we have the following evidence:
> > > > >
> > > > > - Our new Figure 5 shows that a large variety of pairs of models (A,B) reach the same average reconstruction accuracy measured in SSIM locally within the region containing the pathologies.
> > > > >
> > > > > - Figure 17 shows that the pathologies and the reconstructions from network A and B are nearly indistinguishable.
> > > > >
> > > > > - We now added Figure 18 which shows a scatter plot of the reconstructions by network A and B for VarNet for each pathology individually. The plot shows that there is a larger variance across SSIM values (showing that as the reviewer mentioned "some images are easier to reconstruct than others"). However, how well A does for a given pathology is very strongly linearly correlated with how well B does on the same pathology. This plot contains all information about the relative performance on individual pathologies, including the tails, as requested.
> > > > > We hope that this resolved issue (ii).
> > > > >
> > > > > Regarding (i), our choice of confidence intervals:
> > > > > Confidence intervals are meant to quantify the variability of the outcome of an experiment; in our paper we compare situations where a network is trained on one dataset A and another network is trained on a dataset B and we are interested in how the two network's performance compares. Thus, the variability arises by re-sampling the dataset and re-training the network, and thus we quantify this uncertainty as a response to the reviews. We find that those confidence intervals are very small. We think that this is the right way to quantify the uncertainty in our setup.
> > > > >
> > > > > Contrary, the confidence interval in the FastMRI paper the reviewer mentioned above ('as large as 0.07') are over the SSIM values (for example, say there are 10 images in the test sets, then each image has a different SSIM value and the confidence intervals quantify the variability over those values). This measures how the SSIM values vary across images in the test set. Since we have a large variety of test sets that we use in the paper, this number would mainly communicate the diversity of the test-set, e.g., test sets with brains and knees would have much larger variability than test sets with knees and brains separately. See the new Figure 18 on the spread of the SSIM values for one example.
> > > > >
> > > > > Adding those confidence intervals to our plots (e.g., to Figure 3) would be highly confusing as the intervals would be relatively large (since as the reviewer puts it  "some images are easier to reconstruct than others"), even if the uncertainty about the mean (i.e., the bars in Figure 3) would be very low. Even if  trained on a dataset containing millions of images, the confidence intervals would not get any smaller for example. We note that our choice of confidence intervals is standard practice in the machine learning community, e.g., to quantify the uncertainty of classification we do not measure the variance of the classifier across images, instead we give confidence intervals over the test-set-size, quantifying uncertainty when we repeat the experiment.
> > > > >
> > > > > We are happy to add this explanation for our choice of confidence intervals to the paper, and hope this resolved the concern.

---

### Official Review · Reviewer_xWg5 · 2023-11-01

**Soundness:** 3 good
**Presentation:** 4 excellent
**Contribution:** 4 excellent
**Rating:** 10
**Confidence:** 4

**Summary:**

This paper performs a thorough empirical evaluation of the effect of variability in the acquired training data on the in- and out-of-distribution performance of deep learning models trained for MRI reconstruction. The paper contains experiments supporting several points:

- Training a single model on two different distributions yields similar in-distribution performance as two models trained separately on the different datasets.
- Training a single model on multiple distributions improves out-of-distribution performance (though it does not match performance of a model trained specifically for the new distribution).
- A model trained on healthy subjects can generalize well to subjects with pathologies, even when the model has never seen pathologies during training.
- “Distributional overfitting” can occur where out-of-distribution begins to decrease while in-distribution performance continues to increase.

Based on the observed trends, the paper demonstrates that a single model trained on a large collection of datasets provides better out-of-distribution performance and comparable in-distribution performance to networks trained solely on FastMRI.

**Strengths:**

- The paper contains extensive experiments across various different data splits (based on anatomy, contrast, field strength, pathology, and data source) and convincingly demonstrates consistent trends in in-/out-of-distribution performance across many of these splits and across many architectures. The experiments are carefully done (for example, in Section 3/Figure 2, reporting results not just on the conglomerate dataset, but also on a subset of the data whose size roughly matches the sizes of P/Q).
- The questions studied in this paper regarding generalization across scan parameters and pathology state are very important in the context of medical imaging, where data is hard to come by and varies significantly from site-to-site. The paper provides actionable insights for deep learning practitioners (for example, when training on FastMRI, it is good to include data both with and without fat suppression).
- The paper is clearly written and the graphics are largely well-designed to distill the conclusions for the reader. For example, figure 7 is a somewhat unconventional data visualization but makes the point very clearly.

**Weaknesses:**

I will stress that I think this is a very strong paper and that I don’t believe that the weaknesses below are reasons to reject the paper.
- **Limited tuning of hyperparameters.** From Appendix D, it appears that all models are trained with the exact same hyper parameters. A more thorough version of this experiment would tune these hyperparameters for each architecture and dataset. However, I recognize that, given the number of models trained in the paper, this would be exceptionally computationally expensive. The chosen hyperparameters seem to be fairly standard values that a deep learning practitioner might use to initialize a model, and the observed trends are largely consistent across architecture, dataset split, etc, so I am okay with the current experimental setup — but it may be good to include a note about this in the Limitations section.
- **Limited discussion of model capacity.** Related to the above point, I am particularly curious whether the observed trends hold for smaller models. This is motivated by some informal experiments I did on FastMRI data with/without fat suppression with much smaller models than those in this paper, where I observed that including both data splits did decrease model performance compared to models trained separately on each split. One hypothesis could be that models with lower capacity are less able to capture the variability in multiple data splits, leading to this effect. The bigger models in this paper definitely achieve better performance overall and thus are the right ones to report results for here, so I don’t expect the current experiments to be expanded to cover this point and think the paper is strong enough as is. But if the authors have already conducted some related experiments, they may be useful to include in the appendix, and/or this may be another point for the discussion.
- **Limited utility of the early stopping criteria for distributional overfitting.** Unlike traditional overfitting, where the early stopping epoch can be chosen by looking at a held-out subset of the current dataset, it seems to be much harder to monitor early stopping for the distributional overfitting case. Because we are interested in performance on out-of-distribution data, in a realistic scenario, we likely don’t have the out-of-distribution data to monitor in the first place (if we did, we’d want to include it in the training dataset). I see in section 7 that early stopping was performed by tracking validation loss on the fastMRI knee validation set, which seems like an “in-distribution” loss to me, because the fastMRI knee training set was included in the training data. So it is not clear to me to what extent distributional overfitting is being properly avoided here. I don’t think this is a major limitation; even as is, the learned model outperforms the baselines on the out-of-distribution tests. But this would be a good point to discuss in section 6.
- **It is hard to track the same model across some figures.** In Figures 4 and 5, multiple versions of the same architecture are reported within the same figure, trained on different splits of the data. In the current data visualization in Fig 5 for example, it is hard to track which pair of pentagons correspond to the same model configuration. I wonder if it would be useful to have an additional graphic plotting the *difference* in SSIM on Q for each *difference* in SSIM on P, with one datapoint for each model. This would partition the points into four quadrants showing relative improvement/performance decrease on P and Q, and a compelling result would be to show that no datapoint shows a dramatic decrease in performance on Q when trained on just P, as opposed to being trained on P+Q.
- **Confidence intervals are missing in Figs 2-3.** I understand these would clutter some of the other visualizations, but in Figs 2-3 it would be great to see confidence intervals for the bar plots to contextualize the differences between the bars.

**Questions:**

- How do different model hyperparameters (especially pertaining to model size/model capacity) affect these trends?
- How can one effectively choose an epoch for early stopping to avoid *distributional* overfitting?
- Please see the data visualization suggestions above.

---

> ### Author Response · Authors · 2023-11-19
>
> Many thanks for the review and the positive feedback.
>
> - The reviewer has raised a valid and important concern regarding the absence of confidence intervals in Section 3, where we demonstrate that training a model on two distributions yields comparable performance to training on individual distributions. In the revised version, we incorporated mean and standard deviations over 5 independent runs over different data and starting from a different initialization in Figure 3 and we introduced a new Figure 14. From Figure 14, it can be seen that if the dataset size is beyond 3k, the confidence interval is vanishingly small (not visible) and thus we did not add confidence intervals throughout, since this wouldn't be justified given the cost of thousands of GPU hours and the fact that the confidence intervals would be very small.
>
> - The reviewer emphasized the need for a discussion on model capacity and hyperparameter tuning. To address this, we have conducted experiments akin to those in Section 3 but utilizing a much smaller U-net, see the new Appendix E.1. The results align with those obtained with the larger U-net. Furthermore, we describe the process of determining hyperparameters in the new Appendix D. We thank the reviewer for suggesting this.
>
> - We confirm that the reviewer has correctly observed that we track the in-distribution loss as an attempt to mitigate distributional overfitting. This decision is based on the results in Section 6 which suggest that distributional overfitting begins when training is close to convergence.
>
> - Concerning the reviewer's remarks on visualization, we appreciate the interesting suggestion. The current choice of visualization method aligns with the conventions established in the effective robustness literature (see R. Taori et al., 2020, and T. Nguyen et al., 2022). This decision was made to maintain consistency and facilitate comprehension for readers who are already familiar with the effective robustness framework.
>
> References:
>
> R. Taori et al. “Measuring Robustness to Natural Distribution Shifts in Image Classification.” In Advances in Neural Information Processing Systems, 2020.
>
> T. Nguyen et al. “Quality Not Quantity: On the Interaction between Dataset Design and Robustness of CLIP.” In Advances in Neural Information Processing Systems, 2022.

---

> > ### Comment · Reviewer_xWg5 · 2023-11-23
> > **Keeping my original strong accept score + my response to other reviewers recommending rejection**
> >
> > Thanks to the authors for their thorough responses to all reviewers. As noted in my original review, I think this paper should be accepted and keep my original rating.
> >
> > -----
> >
> > I've also read through the discussion with other reviewers (since there is high variance in the scores on this paper). Reviewers 2hGx and YEQb provided scores that tend toward accepting the paper, while reviewers JsA6 and Eeae recommend rejection. While both reviewers JsA6 and Eeae raised helpful points in the discussion, my view is that their objections are not strong enough to reject the paper.
> >
> > **Reviewer JsA6**'s first concern is a lack of confidence intervals and/or tail analysis for the results in Section 3. Like reviewer JsA6, I also noted in my review that confidence intervals _summarizing variance over patients_ would strengthen the analysis, and I further agree that t-tests and a tail analysis would improve the paper. However, I disagree that the lack thereof are reasons to reject the paper; the results already are strong enough to provide useful intuition and insights for practitioners. For example, suppose in the worst case that a t-test showed there _was_ a significant difference between a model trained on (P+Q) and a model trained only on P or only on Q. Even if this were the case, we can tell from the plots in Fig 2 that the effect size is small -- based on these results, a reasonable practitioner may very well decide to train just one model on P+Q and accept a small performance decrease instead of training separate models on P and Q. This choice would be further justified considering that there are several different variables that the data could be split along.
> >
> > Reviewer JsA6's other concern is that SSIM scores may be a misleading metric for small pathologies. This is an interesting point and I was unaware of the work in Radmanesh et al, which is good context. However, while clinician grading of the reconstructed images would of course be the gold standard for evaluating these reconstructions, I don't expect the authors to obtain these labels, which can incur significant monetary cost to generate. This is especially true for a venue like ICLR where the models are not intended for clinical deployment. In the absence of those labels, the high SSIM scores of the overall images and the local regions (from the rebuttal) provide strong evidence of the high quality of the reconstruction, and I think it is better to share those results with the community with very explicit limitations than to reject the paper and prevent practitioners from using the insights from this quite thorough empirical study.
> >
> > **Reviewer Eeae**'s concerns are with a lack new theory and experimental novelty. My view is that theoretical insight is not necessary for an ICLR paper if it provides actionable/practical empirical evidence, which this paper does. Further, I think the authors have accurately summarized the new insights from this work relative to the previous work raised by the reviewer: their experiments extend to out-of-distribution robustness on networks trained with realistic acquired k-space data off the MRI scanner, unlike previous works.

---

### Official Review · Reviewer_YEQb · 2023-11-01

**Soundness:** 3 good
**Presentation:** 4 excellent
**Contribution:** 3 good
**Rating:** 8
**Confidence:** 3

**Summary:**

This paper investigates the impact of the training data on the model’s performance and robustness for image reconstruction of accelerated MRI by conducting diverse experiments. First, they split dataset into two different pairs of distributions including image contrasts (FLAIR, T1 etc.), magnetic fields (1.5T or 3T), and different anatomies (brain or knee), and compared the in-distribution performance of a single model trained on both against individual models trained on each dataset. Also, they evaluated the models on out-of-distribution data regarding image contrasts, magnetic fields, anatomies, and presence of malignancy (training on data from healthy patients and testing on data from non-healthy patients). Also, they found that “distributional overfitting” occurs when training for long, performance on in-distribution data continues to improve marginally while performance on out-of-distribution data abruptly drops. Based on the experiments, the authors claim that using various distributions of training data provides a more robust model compared to developing separate models for individual distributions.

**Strengths:**

-	The topic they investigated should be simple and interesting to researchers like the reviewer in medical imaging machine learning. This is because they have been always curious about whether models trained on a variety of datasets perform better on out-of-distribution test sets than models trained on individual datasets.
-	Diverse and well-designed experiments were conducted to strengthen their claims.
-	When reading through the experiments, the reviewer thought about the following questions based on the previous experiments, but the authors conducted the experiments that answer to my questions. It seems they spent much time on designing experiments for demonstrating their claims.
-	Beyond the explanations of the empirical results, they leveraged the findings to show the benefit of using early stopping onto reducing “distributional overfitting” and improving model’s robustness without compromising in-distribution performance.

**Weaknesses:**

-	What the authors want to claim should be interesting to researchers in this field. However, it would be much better to show another medical imaging application in addition to image reconstruction, like cancer detection, lesion segmentation etc.
-	According to Table 1, the datasets seem diverse in terms of “View” and “Vendor” as well. So, it would be much more interesting to conduct experiments considering the two factors.
-	Also, it seems they synthesized accelerated MRI by under-sampling the fully sampled k-space from the original MRI images. Then, it would be interesting to compare models trained on k-space data sampled with different frequency (eg. 4-fold vs 8-fold)

**Questions:**

-	All experiments the authors did are well described and their results make sense to the reviewer. So, I’d like to give the authors suggestions about the experiments like listed in “Weaknesses”.

---

> ### Author Response · Authors · 2023-11-19
>
> Many thanks for the review, for acknowledging the careful design of our experiments, and for noting that our findings are useful for researchers in medical imaging machine learning.
>
> - The reviewer suggests that extending the current scope of our work to include other medical applications like medical image segmentation would be beneficial. We agree that a data-centric study on improving robustness is also important in other medical applications like detection and segmentation, but since those problems and datasets are quite different from image reconstruction, they are unfortunately beyond the scope of this paper. We did, however, add more experiments on other image reconstruction networks like VarNet and ViT to further strengthen our claims.
>
> - The reviewer would like to see experiments regarding distribution-shifts with respect to vendors and anatomic views. In Section 7 of the original manuscript we have the mentioned distribution-shifts, however they were hidden since we did not point out that there are those shifts. For example, the distribution shift from fastMRI knee to NYU dataset is a distribution shift in anatomic views, and from fastMRI to Stanford 2D or CC-359 is a distribution shift related to vendors. We have updated the corresponding paragraph in Section 7 of the revised manuscript to emphasize that those shifts are in terms of vendors and anatomic views.
>
> - The reviewer would like to see experiments beyond 4-fold acceleration. While it could be interesting to see what happens at higher acceleration rates, we chose the 4-fold acceleration setup to provide more practical insights, as exceeding 4-fold acceleration tends to diminish clinical relevance (M. Muckley et al., 2021; A. Radmanesh et al., 2022). We added the explanation to our revised manuscript in Section 2.
>
> References:
>
> M. Muckley et al. “Results of the 2020 fastMRI Challenge for Machine Learning MR Image Reconstruction.” IEEE Transactions on Medical Imaging, 2021.
>
> A. Radmanesh, et al. "Exploring the Acceleration Limits of Deep Learning Variational Network–based Two-dimensional Brain MRI." Radiology: Artificial Intelligence, 2022.

---

### Official Review · Reviewer_2hGx · 2023-11-11

**Soundness:** 3 good
**Presentation:** 4 excellent
**Contribution:** 2 fair
**Rating:** 6
**Confidence:** 5

**Summary:**

Deep learning models tend to overfit the input data distribution, which has been shown and studied by several papers in the literature before. This paper studies the effects of input distributional shifts (due to anatomy brain vs knee, contrast FLAIR vs T1, and Magnetic field 3T vs 1.5T) on deep learning models' performance for the task of accelerated MRI reconstruction. The authors propose training on diverse datasets as a solution to increase the robustness of deep learning models to the aforementioned shifts. The authors designed the study problem methodically by first showing that training on diverse datasets isn't worse than training on a single dataset. Then they extend the idea to multiple datasets and show the results on out-of-distribution (held out datasets, not shown during training) datasets. The results indicate that training on diverse datasets is better for out-of-distribution generalization for Accelerated MRI reconstruction.

**Strengths:**

- The paper is well-written and easy to read.
- The experimental design is reasonable and logical for studying input distribution shifts.
- Reporting results on multiple shifts (due to anatomy brain vs knee, contrast FLAIR vs T1, and Magnetic field 3T vs 1.5T), shows that problems exist for several variations of input distribution shifts. Also puts the proposed solution in a better standing.
- Reporting results on diverse datasets (16 used in the paper) shows the general applicability of the inferences made.

**Weaknesses:**

- The paper does not explore vendor distribution shifts, raising questions about the model's generalizability across different MRI vendors.

- The paper focuses on input-distribution shift in visual aspects of MRI images but does not delve into how these shifts manifest in k-space data, the raw data format for MRI.

- The comparison between Figures 1 and 2 suggests that increased in-distribution data size correlates with better out-of-distribution performance. However, the paper does not explicitly study the impact of dataset size on model performance with out-of-distribution data.

- The study utilizes full datasets from different sources for training, but it doesn't explore whether using a subset of these datasets could be equally effective.

- The paper does not adequately discuss its limitations.

**Questions:**

- Is there a reason vendor distribution shift was not studied? Will a model trained using data for GE vendors work for Siemens?

- The premise of the paper is based on input-distribution shift. While visually different contrast images, anatomy, etc have distribution shift in the data. However, the input to MRI reconstruction networks is k-space data (subsampled MRI). How much of the data shift is actually present in k-space representation? Is the k-space representation for different contrast/anatomy/Magnetic-field images actually different distributions not clear in the paper? Studying histograms of amplitude and phase information might give an answer to this question. The distribution shift in the visual domain vs in the Fourier(k-space) domain might be completely different.

- Comparing results from Figures 1 and 2, It looks like if the in-distribution data size increases (1.3k vs 20k, 400 vs 6k), it also shows an increase in out-of-distribution performance (SSIM on Q is always higher for a larger dataset in Figure 1). A study of the impact of dataset size increase vs out-of-distribution performance is also warranted. This will support the idea that add datasets from diverse resources is better than increasing the dataset size of a single source.

- The study has used full datasets from different resources while training. Is there a need to use all samples from different datasets? Maybe you only need a few to inform the model of variation in input distribution with a few samples. Maybe P+Q is not necessary. P+0.005Q is sufficient for good model training ?

- The paper lacks a discussion on the limitations of the results. For example, this is only applicable in cases where data from different resources are available at training time. Other limitations might also include the amount of training time required to train the model with a large dataset size. Comparison of training times for different models might also be a relevant metric to the study.

---

> ### Author Response · Authors · 2023-11-19
>
> Many thanks for the feedback and for noting that the study is methodologically designed, well written, and that our main finding that training on diverse datasets improves out-of-distribution generalization rests on a large variety of experiments.
>
> - Regarding that we do not 'explicitly study the impact of dataset size on model performance with out-of-distribution data': Section 4 and 5 contain experiments with different dataset sizes, but we indeed do not consider different training set sizes throughout for the following reason: Training on larger datasets leads to better models, but not necessarily more robust models. Specifically, in and out-of-distribution performance is correlated for the distribution shifts we consider, see [3] and Figure 4 in our paper, and see [1,2] for correlations of in-and out-of-distribution relations in classification and beyond. In Figure 4, increasing the training set size improves a model in that it improves both in and out-of-distribution performance (it moves on the line) but it is not more robust than expected of a model with a given in-distribution-performance. Contrary, training on more diverse data, leads to an \emph{effective} robustness gain in that it gives better out-of-distribution performance relative to the in-distribution performance.
>
> - Regarding results for vendor-related distribution shifts: In Section 7 of the original manuscript we consider vendor shifts. Specifically, in Section 7 we introduce distribution-shifts from the fastMRI dataset, which are obtained by Siemens scanners, to the Stanford 2D or CC-359 dataset, which are obtained by GE scanners. In both cases we find that training on a diverse dataset improves robustness. We have updated the corresponding paragraph in Section 7 in the revised manuscript to emphasize this distribution-shift.
>
> - The reviewer states that we do not adequately discuss the limitations of our results and further suggests that we discuss the training times of our models: In the revised manuscript we have expanded the discussion on limitations by adding that in practice it might be difficult or expensive to collect diverse and large datasets. We believe that the training time is not a significant concern, given that the current bottleneck in medical imaging is data availability rather than computational resources. Nevertheless, we agree that stating the training times is important, and we have included this information in Appendix D of the revised manuscript.
>
> - Regarding distribution-shifts related to visual aspects of MRI images vs. how these shifts manifest in k-space data: Since k-space and image are related through the Fourier transform, a distribution shift in the image space is also a distribution shift in the k-space.
>
> - The reviewer asks if mixing data from two distributions, P and Q, in a disproportionate manner like ‘P+0.005Q is sufficient for good model training’: In terms of in-distribution performance, we find that skewed data up to 90%P and 10%Q do not pose a problem (see Figure 3 in the revised manuscript). In case of distribution-shifts from distribution P to distribution Q, we do not have data from distribution Q available. If data from distribution Q was available, then including it into the training set would eliminate the distribution shift.
>
> References:
>
> [1] B. Recht et al. “Do ImageNet Classifiers Generalize to ImageNet?” In International Conference on Machine Learning, 2019.
>
> [2] R. Taori et al. “Measuring Robustness to Natural Distribution Shifts in Image Classification.” In Advances in Neural Information Processing Systems, 2020.
>
> [3] M. Darestani et al. “Measuring Robustness in Deep Learning Based Compressive Sensing.” In International Conference on Machine Learning, 2021.

---

> > ### Comment · Reviewer_2hGx · 2023-11-22
> >
> > Thanks to the authors for thorough response to the concerns I raised.  After careful consideration of your responses and in light of other reviews, I have decided to maintain my original score.

---

### Author Response · Authors · 2023-11-19
**Summary of changes in the paper**

We thank all the reviewers for their detailed feedback. In the following we summarize the major changes in the revised version of the paper.

- To address the concerns of Reviewer JsA6We and Reviewer xWg5 regarding the absence of confidence intervals, we have made following changes: We incorporated mean and standard deviations over 5 independent runs over different data and starting from a different initialization in Figure 3 and we introduced a new Figure 14. From Figure 14, it can be seen that if the dataset size is beyond 3k images, the confidence interval is vanishingly small, thus we didn't include confidence intervals throughout (they would not be visible and very expensive to obtain).

- Regarding Reviewer Eeae27’s concern that we didn’t study models beyond the U-net in certain experiments: We have now included results for the End-to-end VarNet to Section 3 (see Figure 2) and Section 7 (see Figure 8), and observe the same qualitative results as with the U-net. Moreover, we also included results for the ViT (see Figure 8 and 13) to demonstrate that our results continue to hold true even for non-convolutional architectures.

- Regarding Reviewer JsA6We’s concern that global SSIM scores for evaluating pathology reconstructions (Section 5) could be problematic due to small pathology sizes: In the revised paper, we have re-evaluated the models only on the region containing the pathology (instead of computing metrics on the entire image), see the revised Figure 5. We further distinguished between small pathologies and relatively large pathologies. The results indicate that models trained on data without pathologies can reconstruct pathologies as accurately as models trained on data with pathologies, regardless of the pathology size.

- Addressing Reviewer 2hGx's concern about a more comprehensive discussion of limitations, we have expanded the discussion on limitations by adding that in practice it might be difficult or expensive to collect diverse and large datasets. Moreover, upon the suggestions from Reviewer 2hGx and Reviewer xWg5, we have included information regarding training time and hyperparameter selection in Appendix D.

---

### Meta-Review · Area_Chair_AAXN · 2023-12-17

**Metareview:**

This paper studies the effects of input distributional shifts (due to anatomy brain vs knee, contrast FLAIR vs T1, and Magnetic field 3T vs 1.5T) on deep learning models' performance for the task of accelerated MRI reconstruction. The authors demonstrate that training on diverse datasets increases the robustness of deep learning models to the distribution shifts. The authors first show that training a single model on diverse datasets is as good as training separate models on separate dataset. Then they show the results on out-of-distribution (held out datasets) datasets. The results indicate that training on diverse datasets is better for out-of-distribution generalization for Accelerated MRI reconstruction.

Strengths:
+ The paper is well-written and easy to read.
+ The experimental design is reasonable and logical for studying input distribution shifts.
+ The experiments report results on different models (e.g., U-Net and ViT).
+ The problem of identifying and mitigating distributional shifts for deep learning accelerated MRI is of significant real-world relevance. It is critical for building a trustworthy deep learning driven MRI reconstruction system.
+ The questions studied in this paper regarding generalization across scan parameters and pathology state are very important in the context of medical imaging

Weaknesses
- Training a model with "diverse" datasets enhances the robustness. This is an interesting empirical result for the datasets tested by authors, but this conclusion is widely known.
- Two  reviewers expressed concerns about the conclusions drawn by the authors based on the experiments. I have some similar reservations as well. For example
    - the paper primarily used 4x acceleration factor for all the experiments. The reported SSIM for all the experiments is close to 0.9 and variations for different experiments is within +/- 0.03. One concern is that with this SSIM range all the reconstructed images are almost the same. While the bar plots in Fig. 2,3 visually show a gap between P vs Q vs P+Q, the absolute difference is very small.
    - Fig. 7 illustrates the effects of early vs late stopping and authors conclude the out-of-distribution increases then drops. If we notice the y-axis, the variation is within a small interval inside [0.775, 0.78]. With such a small change, I find it difficult to draw any conclusion.
- The paper did not separate the effects of out-of-distribution samples and complexity of the problem. For instance, 3T measurements generally provide higher SSIM compared to 1.5T. We see this effect in Fig. 2,3,4 that models trained for 1.5T perform well on 3T measurements.
- The paper presents an interesting observation that the network trained without pathological cases can correctly recover pathological cases. I agree with the reviewers that this statement can be misleading and a deeper analysis is needed to substantiate the claim that models trained on healthy subjects can reconstruct pathology. My impression is that the inverse problem is simple enough that the network can recover images with high fidelity for most of the cases. This effect is further supported by the large SSIM values of the reconstruction.
- The so-called data diversity is a vague term that authors did not define explicitly. The authors seem to infer that if the model trained on P performs well on Q, then P is diverse. This reasoning is not very satisfactory.

**Justification For Why Not Higher Score:**

Several factors contributed to this recommendation. I would have recommended a higher score if some or all of these issues were resolved.
1. All the results reported in the paper have large SSIM values (close to 0.9) and the differences are within +/- 0.03. This makes it difficult to ascertain what role the so-called diversity play and whether performance actually change with in-distribution vs out-of-distribution data.
2. The observations made in the paper that a diverse training data provides better results compared a restricted dataset are generally known. This paper does not add anything significantly new in that regard.
3. The so-called data diversity is a vague term that authors did not define explicitly. The authors seem to infer that if the model trained on P performs well on Q, then P is diverse. This reasoning is not very satisfactory.

**Justification For Why Not Lower Score:**

N/A

---

### Decision · Program_Chairs · 2024-01-16

Reject